# Streaming Autoregressive Video Generation via Diagonal Distillation

**Jinxiu Liu[1,*]  Xuanming Liu[2,*]  Kangfu Mei[3]  Yandong Wen[2]  Ming-Hsuan Yang[4]  Weiyang Liu[5]**

[1]South China University of Technology   [2]Westlake University   [3]Johns Hopkins University
[4]University of California, Merced   [5]The Chinese University of Hong Kong

**SphereLab.ai/diagdistill**

## Abstract

Large pretrained diffusion models have significantly enhanced the quality of generated videos, and yet their use in real-time streaming remains limited. Autoregressive models offer a natural framework for sequential frame synthesis but require heavy computation to achieve high fidelity.Diffusion distillation can compress these models into efficient few-step variants, but existing video distillation approaches largely adapt image-specific methods that neglect temporal dependencies. These techniques often excel in image generation but underperform in video synthesis, exhibiting reduced motion coherence, error accumulation over long sequences, and a latency-quality trade-off. We identify two factors that result in these limitations: insufficient utilization of temporal context during step reduction and implicit prediction of subsequent noise levels in next-chunk prediction (*i.e.*, exposure bias). To address these issues, we propose *Diagonal Distillation*, which operates orthogonally to existing approaches and better exploits temporal information across both video chunks and denoising steps. Central to our approach is an asymmetric generation strategy: more steps early, fewer steps later.This design allows later chunks to inherit rich appearance information from thoroughly processed early chunks, while using partially denoised chunks as conditional inputs for subsequent synthesis. By aligning the implicit prediction of subsequent noise levels during chunk generation with the actual inference conditions, our approach mitigates error propagation and reduces oversaturation in long-range sequences. We further incorporate implicit optical flow modeling to preserve motion quality under strict step constraints.Our method generates a **5-second video** in **2.61 seconds** (up to **31 FPS**), achieving a **277.3×** speedup over the undistilled model.

## 1 Introduction

Recent years have witnessed the rapid progress of diffusion models in video generation. A major enabler of such progress has been Diffusion Transformer architectures (Peebles & Xie, 2023), which leverage bidirectional attention to denoise all video frames simultaneously (Blattmann et al., 2023a;b; Brooks et al., 2024; Kong et al., 2024; Polyak et al., 2024; Villegas et al., 2022; Wan et al., 2025; Yang et al., 2024). While effective for offline generation, this design requires the entire video to be generated at once, as each frame can attend to all others, including future ones. As a result, such models face fundamental limitations in real-time applications, including game simulation (Deng et al., 2024; Peebles & Xie, 2023; Song et al., 2023; Vondrick et al., 2016) and robot learning (Ge et al., 2022; Jolicoeur-Martineau, 2018; Wang et al., 2023), where future frames are unavailable when generating the current frame.

Autoregressive (AR) models are well-suited for streaming video generation, as their chunk-by-chunk synthesis naturally aligns with real-time constraints (Bruce et al., 2024; Kondratyuk et al., 2023; Ren et al., 2025; Wang et al., 2024; Weissenborn et al., 2019; Yan et al., 2021). However, traditional GPT-style models (Wang et al., 2024; Yan et al., 2021) often suffer from limited visual quality (Gao et al., 2024a). To address this, recent works (Jin et al., 2024; Weng et al., 2024; Teng et al., 2025) integrate diffusion processes into AR generation. Yet these methods still require multiple denoising steps per

---
*Equal contribution with alphabetical order.

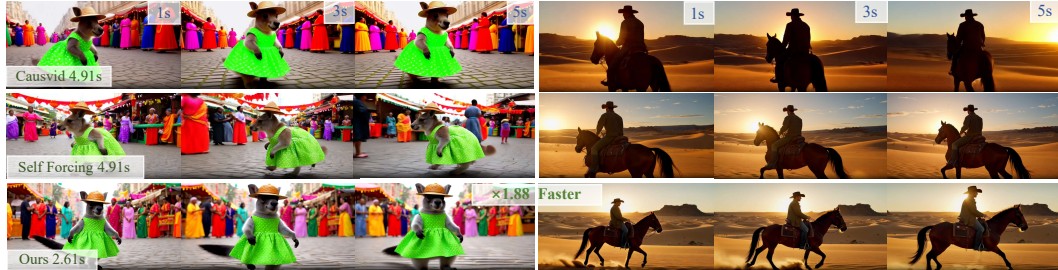

Figure 1: Diagonal Distillation achieves comparable quality to the full-step model while significantly reducing latency. The method yields a 1.88× speedup on 5-second short video generation on a single H100 GPU.

chunk, which hinders real-time deployment. To reduce inference latency, step distillation (Yin et al., 2025; Huang et al., 2025; Yin et al., 2024b) has been introduced to distill multi-step diffusion models into efficient few-step sampling AR model. Recent training methods (Chen et al., 2024; Gao et al., 2024b; Gu et al., 2025; Hu et al., 2024; Li et al., 2024b; Liu et al., 2024b; Weng et al., 2024; Yin et al., 2025; Zhang et al., 2025a;b) have further improved stability and efficiency, making interactive applications increasingly feasible (Arriola et al., 2025; Liu et al., 2024c).

Despite the encouraging advancement, existing video distillation methods are largely adapted from image generation, and their direct extension to video often yields suboptimal results. This limitation arises from insufficient consideration of the temporal dimension and the neglect of inter-frame consistency. As a result, multi-step sampling remains essential for maintaining high-quality video generation. For example, while autoregressive frameworks such as Causvid (Yin et al., 2025) and Self-Forcing (Huang et al., 2025) can reduce latency, they still require multiple steps per segment, and compressing them to fewer steps leads to noticeable performance degradation.

Our guiding insight is that, in autoregressive video generation, predicting the next chunk inherently requires predicting the next noise level (see Figure 2). This implicit prediction, however, introduces two critical challenges. First, autoregressive video models often suffer from exposure bias. When predicting the next chunk conditioned on previously generated clean frames, the model must implicitly predict the next noise level for subsequent frames. This can lead to progressive visual quality degradation, such as over-saturation in later frames, as errors in noise-level prediction can accumulate over time. Although techniques like Self-Forcing (Huang et al., 2025) have been proposed to mitigate exposure bias by using model-generated content during training, they still struggle to maintain visual quality over long sequences. Second, the same phenomenon implies that if structural priors are captured in early chunks, later chunks can generate relatively clear frames even with fewer denoising steps. However, existing distillation approaches often discard valuable temporal context accumulated across denoising steps in video generation models, which is essential for preserving coherence and detail when reducing the sampling steps.

Motivated by these observations, we introduce a flow-aware *diagonal distillation* framework, termed *DiagDistill* that redefines the temporal context incorporation by leveraging information across both time and denoising steps. Different from common practices that process chunks in isolation, our method employs a novel diagonal attention mechanism operating jointly across time and denoising steps. This results in a diagonal denoising trajectory wherein earlier chunks are denoised with more steps, while later chunks use progressively fewer steps. This strategy improves computational efficiency by using less denoising steps in total and allows each chunk to inherit denoising trajectories from prior chunks as contextual priors, which leads to a training paradigm we term *Diagonal Forcing*. By explicitly simulating diagonal denoising paths during training through controlled noise injection, Diagonal Forcing enhances self-conditioned generation and mitigates error accumulation in long videos. However, we empirically observe that employing very few steps in later chunks can attenuate motion amplitude. To address this, we introduce Flow Distribution Matching, which integrates explicit temporal modeling into the distillation loss. This approach preserves dynamic consistency by ensuring the predicted motion distributions align with those of the full-step model, thus ensuring that the student model not only matches the teacher in image quality but also faithfully preserves motion characteristics. We summarize the major contributions of this work below:

- We propose *Diagonal Distillation*, a method for efficient autoregressive video generation. Instead of assigning a fixed number of denoising steps to all chunks, it allocates more steps to earlier

Figure 2: When the training data uses explicit noise frames as conditions in Causvid (Yin et al., 2025), the next chunk prediction essentially functions as an implicit next noise level prediction. We observe that even with single-step prediction, the image progressively becomes clearer.

chunks and progressively fewer to later ones. This strategy leverages contextual structural priors in autoregressive video generation, achieving a good balance between quality and efficiency.

- We introduce *Diagonal Forcing*, built upon diagonal distillation, as a unified method operating across both temporal and denoising-step dimensions. It leverages trajectories from preceding chunks as contextual priors and explicitly simulates diagonal denoising paths during training via controlled noise injection, effectively mitigating long-term error accumulation.

- To address motion degradation and amplitude attenuation in later chunks, we propose *Flow Distribution Matching* that works alongside diagonal distillation. By incorporating explicit temporal modeling into the distillation loss, we find that this approach effectively enhances dynamic consistency and ensures smooth motion transitions.

- Our method achieves state-of-the-art performance on video generation. It generates a 5-second video in 2.61 seconds (up to 31 FPS), reaching a 277.3× speedup over the undistilled model.

## 2 RELATED WORK

**Diffusion Distillation.** Diffusion distillation accelerates sampling via deterministic or distributional approaches. Deterministic methods (e.g., progressive distillation (Salimans & Ho, 2022), consistency distillation (Li et al., 2023; Song et al., 2023), rectified flow (Lamb et al., 2016)) regress noise-to-sample mappings but often yield blurry outputs with few steps due to optimization challenges (Kingma et al., 2021), typically requiring multiple steps for acceptable quality (Li et al., 2023; 2024a). Distributional methods approximate the teacher's distribution using adversarial training (Brooks et al., 2024; Ho et al., 2022), score distillation (Li et al., 2022; Luo et al., 2024), or hybrid objectives. Recent hybrids combine both paradigms but still suffer from one-step artifacts and commonly need multi-step sampling. Representative works include LADD (Sauer et al., 2024a), which relies on expensive pre-generated teacher targets; Lightning (Lin et al., 2024) and Hyper (Ren et al., 2024), which require intermediate timestep supervision; and DMD/DMD2 (Yin et al., 2024b;a) and ADD (Sauer et al., 2024b), which integrate adversarial and score-matching losses. While these distillation methods have shown impressive results in image generation, their direct application to video often yields suboptimal results due to insufficient consideration of the temporal dimension and inter-frame consistency. Our work addresses this problem by proposing a flow-aware diagonal distillation framework specifically designed for video generation, which leverages temporal context across both time and denoising steps to maintain coherence while reducing sampling steps.

**Autoregressive, Diffusion, and Hybrid Video Generation.** Modern video generation is dominated by scalable diffusion and autoregressive (AR) models. Video diffusion models use bidirectional attention to denoise all frames concurrently (Blattmann et al., 2023a;b; Brooks et al., 2024; Deng et al., 2024; Kong et al., 2024; Polyak et al., 2024; Villegas et al., 2022; Wan et al., 2025; Yang et al., 2024), while AR models generate spatiotemporal tokens sequentially via next-token prediction (Bruce et al., 2024; Kondratyuk et al., 2023; Ren et al., 2025; Wang et al., 2024; Weissenborn et al., 2019; Yan et al., 2021; Liu et al., 2025). Hybrid models that merge these two paradigms have recently emerged as a promising direction (Chen et al., 2024; Gao et al., 2024b; Gu et al., 2025; Hu et al., 2024; Jin et al., 2024; Li et al., 2024b; Liu et al., 2024a;b; Weng et al., 2024; Yin et al., 2025; Zhang et al., 2025a;b), also in other sequence domains (Arriola et al., 2025; Liu et al., 2024c). These hybrids typically integrate diffusion into AR generation to boost visual quality, but they still require multiple denoising steps per chunk, hindering real-time deployment. Our work builds on these hybrids, drawing inspiration from Yin et al. (2025) and Huang et al. (2025) to mitigate exposure bias. However, these methods still face challenges with long-term error accumulation and motion degradation when compressed to fewer steps.

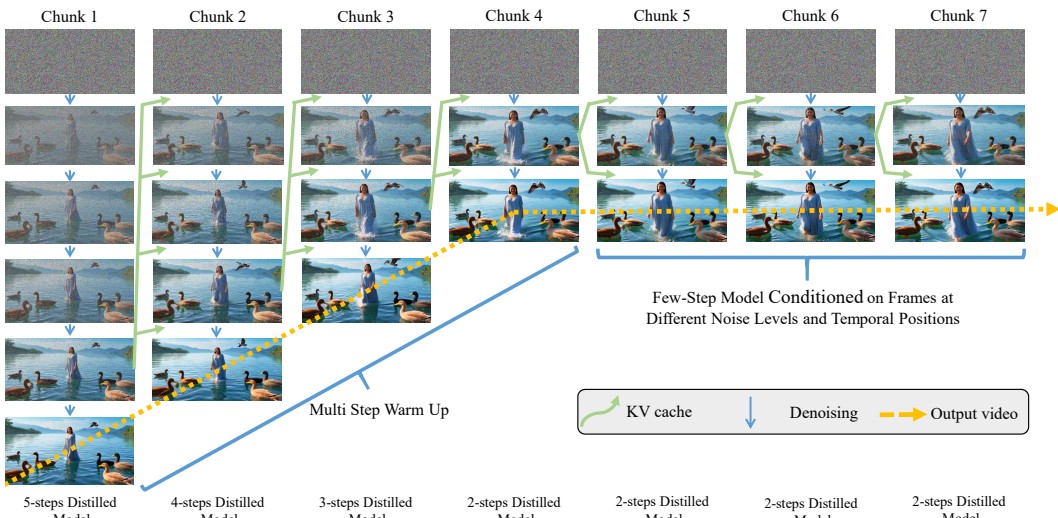

Figure 3: Diagonal Denoising with Diagonal Forcing and Progressive Step Reduction. We give an illustration of our method by starting with five denoising steps for the first chunk and gradually reducing them to two steps by Chunk 7. For chunks with $k \geq 4$, we use a fixed two-step denoising process, reusing the Key-Value (KV) cache from the final noisy frame of the preceding chunk. This design preserves temporal coherence while minimizing latency, and the corresponding pseudo-code is provided in the appendix.

Our diagonal distillation framework addresses these challenges through a novel diagonal attention mechanism that operates jointly across time and denoising steps, achieving efficient computation while maintaining temporal coherence. The diagonal forcing training paradigm simulates diagonal denoising paths to improve self-conditioned generation, while flow distribution matching enforces motion consistency with fewer denoising steps.

## 3 THE DIAGONAL DISTILLATION FRAMEWORK

### 3.1 PRELIMINARY AND FRAMEWORK OVERVIEW

Diffusion Models generate data through an iterative denoising process. The forward diffusion process progressively corrupts a sample $x \sim p_{\text{real}}$ over $T$ steps, such that at timestep $t$, the diffused sample follows $p_{\text{real},t}(x_t) = \int p_{\text{real}}(x)q(x_t|x)dx$, with $q_t(x_t|x) \sim \mathcal{N}(\alpha_t x, \sigma_t^2 I)$, where $\alpha_t, \sigma_t > 0$ are determined by the noise schedule. The model learns to reverse this process by predicting a denoised estimate $\mu(x_t, t)$. The score function of the diffused distribution is:

$$s_{\text{real}}(x_t, t) = \nabla_{x_t} \log p_{\text{real},t}(x_t) = -\frac{x_t - \alpha_t \mu_{\text{real}}(x_t, t)}{\sigma_t^2}. \tag{1}$$

Sampling typically requires many iterative steps. Distribution Matching Distillation (DMD) distills a multi-step diffusion model (teacher) into a one-step generator $G$ by minimizing the KL divergence between the diffused real and generated distributions, $p_{\text{real},t}$ and $p_{\text{fake},t}$. The gradient of this loss is:

$$\nabla \mathcal{L}_{\text{DMD}} = \mathbb{E}_t \left( \nabla_\theta \text{KL}(p_{\text{fake},t} \| p_{\text{real},t}) \right) = -\mathbb{E}_t \left( \int \left( s_{\text{real}}(F(G_\theta(z), t), t) - s_{\text{fake}}(F(G_\theta(z), t), t) \right) \frac{dG_\theta(z)}{d\theta} dz \right), \tag{2}$$

where $z \sim \mathcal{N}(0, \mathbf{I})$, $F$ is the forward diffusion process, and $s_{\text{real}}, s_{\text{fake}}$ are scores from models trained on real and generated data. An additional regression loss is often used for regularization:

$$\mathcal{L}_{\text{reg}} = E_{(z,y)} d(G_\theta(z), y), \tag{3}$$

where $y$ is an image generated by the teacher from $z$. Directly applying DMD to video generation faces a significant challenge: the regression loss $\mathcal{L}_{\text{reg}}$ primarily ensures per-frame quality but fails to explicitly capture the underlying temporal coherence and long-range dependencies between frames, which are critical for video quality. This often results in degraded fluidity and consistency. To overcome this, we extend the DMD framework with two core innovations: 1) a **Diagonal Denoising with Diagonal Forcing** strategy that manages long-sequence generation and reduces error accumulation (Section 3.2); 2) a novel **Flow Distribution Matching** objective that explicitly aligns the temporal dynamics of the student and teacher models (Section 3.3).

## 3.2 Diagonal Denoising with Diagonal Forcing

Building upon the DMD foundation, we present diagonal distillation, a framework for efficient video generation. As illustrated in Figure 3, our approach introduces a Diagonal Denoising strategy that progressively reduces denoising steps across video chunks, combined with a novel Diagonal Forcing mechanism to maintain temporal coherence and mitigate error accumulation.

**Diagonal Denoising: Progressive Step Reduction Strategy.** Our core innovation is a diagonal denoising strategy that allocates computation based on temporal importance. The method assigns more denoising steps to earlier chunks and progressively fewer to later ones, rather than maintaining a constant number of steps across all chunks. This approach achieves an improved trade-off between quality and efficiency by leveraging contextual structured priors in autoregressive video generation. For the first three chunks ($k = 1, 2, 3$), we use distilled models with decreasing steps ($s_k = 5, 4, 3$):

$$\mathbf{X}_k = \mathcal{D}_{s_k}(\mathbf{Z}_k | \tilde{\mathbf{X}}_{<k}), \tag{4}$$

where $\mathbf{X}_k$ is the $k$-th chunk output, $\mathbf{Z}_k \sim \mathcal{N}(0, \mathbf{I})$ is Gaussian noise, and $\tilde{\mathbf{X}}_{<k}$ contains previously noised chunks. For $k \geq 4$, we employ efficient two-step denoising:

$$\mathbf{C}_k = \mathcal{T}(\tilde{\mathbf{X}}_{k-1}), \mathbf{X}_k \quad = \mathcal{D}_2(\mathcal{D}_1(\mathbf{Z}_k | \mathbf{C}_k) | \mathbf{C}_k), \tag{5}$$

where $\mathbf{C}_k$ is the conditioning signal derived from previous chunks, $\mathcal{T}$ denotes the conditioning module, and $\mathcal{D}_1, \mathcal{D}_2$ represent the first and second denoising steps respectively.

**Diagonal Forcing: Contextual Prior Propagation.** The core innovation of Diagonal Forcing lies in its explicit modeling of diagonal denoising trajectories during training through controlled noise injection. This approach ensures temporal coherence across chunks while minimizing error accumulation by conditioning each new chunk on the final noised state from the previous chunk's diffusion process. Specifically, the conditioning input for chunk $k$ is derived from the clean output $\mathbf{X}_{k-1}$ of chunk $k - 1$ through a noise injection operation:

$$\tilde{\mathbf{X}}_{k-1} = \sqrt{\alpha_{k-1}} \cdot \mathbf{X}_{k-1} + \sqrt{1 - \alpha_{k-1}} \cdot \boldsymbol{\epsilon}, \quad \boldsymbol{\epsilon} \sim \mathcal{N}(\mathbf{0}, \mathbf{I}) \tag{6}$$

where $\alpha_{k-1}$ controls the noise schedule along the diagonal path and $\boldsymbol{\epsilon}$ is standard Gaussian noise. This formulation explicitly maintains the diagonal denoising trajectory $\mathbf{X}_k \to \tilde{\mathbf{X}}_{k-1} \to \mathbf{X}_{k-1}$, where $\tilde{\mathbf{X}}_{k-1}$ serves as the KV cache input for chunk $k$. By propagating these noised representations across chunks, the method effectively leverages denoising trajectories from prior chunks as contextual priors. The diagonal alignment of these trajectories ensures that error accumulation is minimized while preserving long-range coherence in the generated output.

## 3.3 Flow Distribution Matching

Motion attenuation in few-step denoising stems from truncated noise estimation paths. We quantify the temporal distribution mismatch through flow-based divergence:

$$\mathcal{E}_{\text{motion}} = D_{\text{KL}}\left(p_{\text{teacher}}(\mathcal{F}(\mathbf{x}) | \mathbf{x}_t) \| p_{\text{student}}(\mathcal{F}(\mathbf{x}) | \mathbf{x}_t)\right) \tag{7}$$

where $\mathcal{F}(\mathbf{x})$ represents the motion flow field extracted from video sequence $\mathbf{x}$. This measures the distributional divergence between teacher and student in the temporal dimension.

The standard Distribution Matching Distillation (DMD) framework minimizes spatial divergence through reverse KL minimization. We extend this to the temporal domain by defining flow distribution matching:

$$\nabla_\phi \mathcal{L}_{\text{DMD}}^{\text{flow}} \triangleq \mathbb{E}_t\left(\nabla_\phi \text{KL}\left(p_{\text{gen,flow},t} \| p_{\text{data,flow},t}\right)\right) \tag{8}$$

where $p_{\text{data,flow},t} = p(\mathcal{F}(\mathbf{x}) \mid \Psi(\mathbf{x}, t))$ is the smoothed flow distribution from real data, and $p_{\text{gen,flow},t} = p(\mathcal{F}(\mathbf{x}) \mid \Psi(G_\phi(\epsilon), t))$ is the generator's flow distribution. The gradient approximation for flow distribution matching follows the DMD framework:

$$\nabla_\phi \mathcal{L}_{\text{DMD}}^{\text{flow}} \approx -\mathbb{E}_t\left[\int \left(s_{\text{data}}^{\text{flow}}\left(\Psi(G_\phi(\epsilon), t), t\right) - s_{\text{gen},\phi}^{\text{flow}}\left(\Psi(G_\phi(\epsilon), t), t\right)\right) \frac{dG_\phi(\epsilon)}{d\phi} d\epsilon\right], \tag{9}$$

where $s_{\text{data}}^{\text{flow}}$ and $s_{\text{gen},\phi}^{\text{flow}}$ are the flow score functions defined as:

$$s^{\text{flow}}(\mathbf{x}_t, t) = \nabla_{\mathbf{x}_t} \log p(\mathcal{F}(\mathbf{x}) | \mathbf{x}_t). \tag{10}$$

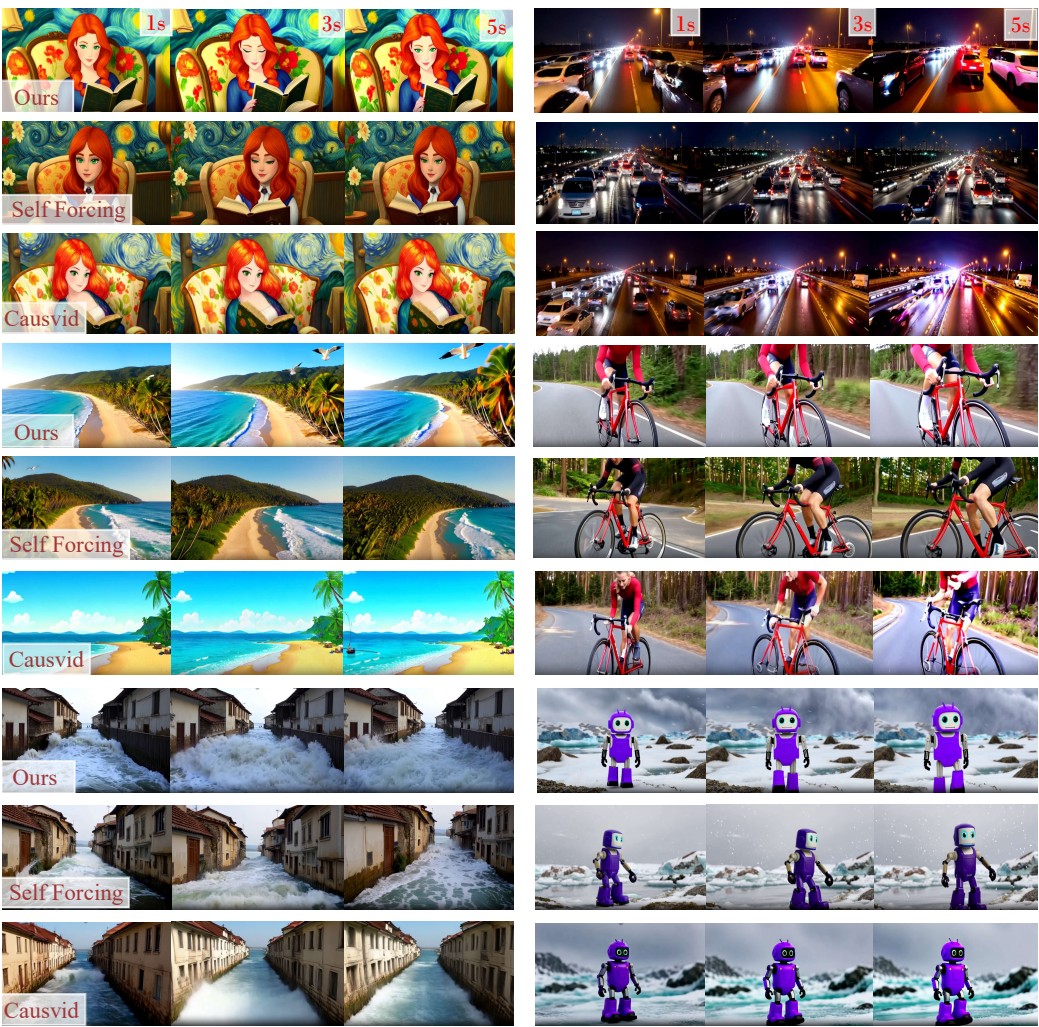

Figure 4: Comparing the results from three different models. For more results, please refer to the Appendices.

where $F(\cdot)$ denotes the flow extractor that maps a video representation to its corresponding motion (optical flow) features.

To operationalize this framework, we employ a flow regression loss for feature alignment:

$$\mathcal{L}_{\text{reg}}^{\text{flow}} = \mathbb{E}_{t,\epsilon} \left[ \| \mathcal{F}(G_\phi^{\text{teacher}}(\epsilon, t)) - \mathcal{F}(G_\phi^{\text{student}}(\epsilon, t)) \|_2^2 \right], \tag{11}$$

where $G_\phi(\epsilon, t)$ denotes the generator output at timestep $t$. Our method uses a lightweight, self-contained motion feature extraction module $\mathcal{F}(\cdot)$ that operates directly on latent representations, avoiding dependencies on external pre-trained optical flow estimators. Specifically, we implement $\mathcal{F}(\cdot)$ as a learnable representation with convolution on latent difference: it first computes the difference between consecutive latent frames, then applies convolutional layers to extract local motion patterns, followed by an MLP for feature adaptation. The student version is trainable with gradient flow, while the teacher components are updated via EMA, ensuring stable and efficient motion representation learning. The overall objective combines both spatial and temporal distribution matching:

$$\mathcal{L}_{\text{Total}} = \lambda_{\text{spatial}} \mathcal{L}_{\text{DMD}}^{(\text{grad})} + \mathcal{L}_{\text{reg}} + \gamma \left( \lambda_{\text{flow}} \mathcal{L}_{\text{DMD}}^{\text{flow},(\text{grad})} + \mathcal{L}_{\text{reg}}^{\text{flow}} \right), \tag{12}$$

where $\gamma$ weights the temporal terms. We set $\lambda_{\text{spatial}} = 4, \lambda_{\text{flow}} = 4$. This framework jointly minimizes motion distribution divergence while maintaining spatial fidelity in the distilled video model.

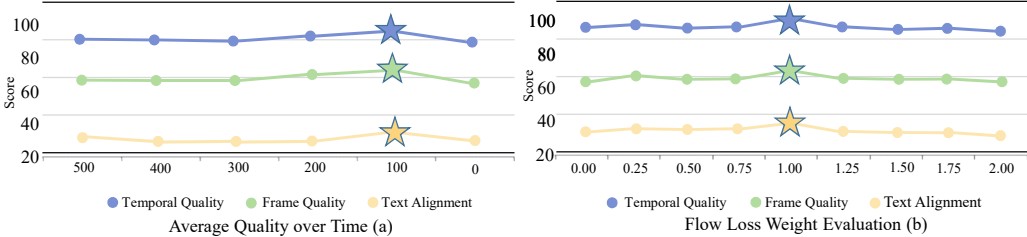

Figure 5: Ablation study results. (a) Performance evaluation across different diagonal forcing timesteps, demonstrating optimal outcomes at 100 steps (1000 steps correspond to complete noise addition, while 0 steps represent the clean frame);(b)Impact of motion loss weight on model performance.

## 4 EXPERIMENTS AND RESULTS

### 4.1 IMPLEMENTATION DETAILS

**Training Details.** We implement DiagDistill using Wan2.1-T2V-1.3B (Wan et al., 2025), a model based on Flow Matching (Lipman et al., 2022) that is capable of generating 5 videos at 16 FPS with a resolution of $832 \times 480$. For both ODE initialization and Diagonal Distillation training, we sample text prompts from a filtered and LLM-extended version of VidProM (Wang & Yang, 2024).

**Inference Details.** To assess real-time applicability, we measured both throughput (frames per second) and first-frame latency, acknowledging that true real-time performance requires exceeding video playback rates while maintaining imperceptible delay. All speed tests were conducted on a single NVIDIA H100 GPU with tiny VAE (Boer Bohan, 2025). The core component is the rolling KV cache mechanism following Self-Forcing (Huang et al., 2025), which operates with a chunk size of 3 frames. Our buffering strategy is implemented using a fixed-size KV cache that maintains context from the most recent 4 chunks, resulting in a consistent memory footprint of 17.5 GB. For detailed ablation analysis please refer to the Appendices.

**Evaluation Details.** We evaluated visual quality and semantic consistency using VBench (Huang et al., 2024). Temporal Quality is the average of Subject Consistency, Background Consistency, Temporal Flickering, Motion Smoothness, and Dynamic Degree. Frame Quality is the average of Aesthetic Quality and Imaging Quality. Text Alignment is the average of Object Class, Multiple Objects, Human Action, Color, Spatial Relationship, Scene, Appearance, Style, and Temporal Style. The aggregation method for each score is a simple arithmetic mean of the normalized scores from its constituent sub-dimensions. This evaluation approach is consistent with prior works like Causvid and Self-Forcing for fair comparison.

### 4.2 COMPARISON WITH STATE-OF-THE-ART METHODS

We start by evaluating DiagDistill against five state-of-the-art video generation methods: Wan2.1 (Wan et al., 2025), SkyReels-V2 (Chen et al., 2025), MAGI-1 (Teng et al., 2025), Causvid (Yin et al., 2025), and Self-Forcing (Huang et al., 2025).

| Model | Throughput↑ | First-Frame Latency ↓ | Speedup | Total↑ | Quality↑ | Semantic↑ |
|---|---|---|---|---|---|---|
| Wan2.1 (Wan et al., 2025) | 0.78 | 103 | 1.0× | 84.26 | **85.30** | 80.09 |
| SkyReels-V2 (Chen et al., 2025) | 0.49 | 112 | 0.91× | 82.67 | 84.70 | 74.53 |
| MAGI-1 (Teng et al., 2025) | 0.19 | 282 | 0.36× | 79.18 | 82.04 | 67.74 |
| Causvid (Yin et al., 2025) | 17.0 | 0.69 | 149.3× | 81.20 | 84.05 | 69.80 |
| Self-Forcing (Huang et al., 2025) | 17.0 | 0.69 | 149.3× | 84.31 | 85.07 | 81.28 |
| **DiagDistill (Ours)** | **31.0** | **0.37** | **277.3×** | **84.48** | 85.26 | **81.73** |

Table 1: Comprehensive comparison of video generation methods

As shown in Table 1, our method achieves a 277.3× speedup over the Wan2.1 baseline while maintaining competitive visual quality (85.26 vs. 85.3). This represents a 1.53× improvement in latency over the previous fastest method, Self-Forcing (149.3×), alongside superior overall performance and semantic consistency . Qualitative results in Figure 4 further demonstrate advantages in temporal consistency, with smoother frame transitions and fewer dynamic artifacts. Visual fidelity improvements are most apparent in complex motions and textures, where baseline methods exhibit blurring or distortion. These findings collectively show that DiagDistill effectively balances the traditional trade-off between generation quality and computational efficiency.

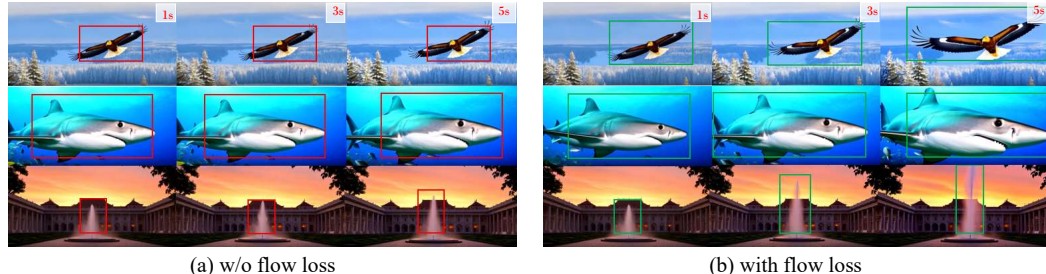

(a) w/o flow loss        (b) with flow loss

Figure 6: Visual comparison of motion effects. (a) Without motion loss shows minimal motion amplitude with only slight object movement; (b) With motion loss demonstrates significantly increased motion amplitude throughout the entire frame, validating our method's effectiveness.

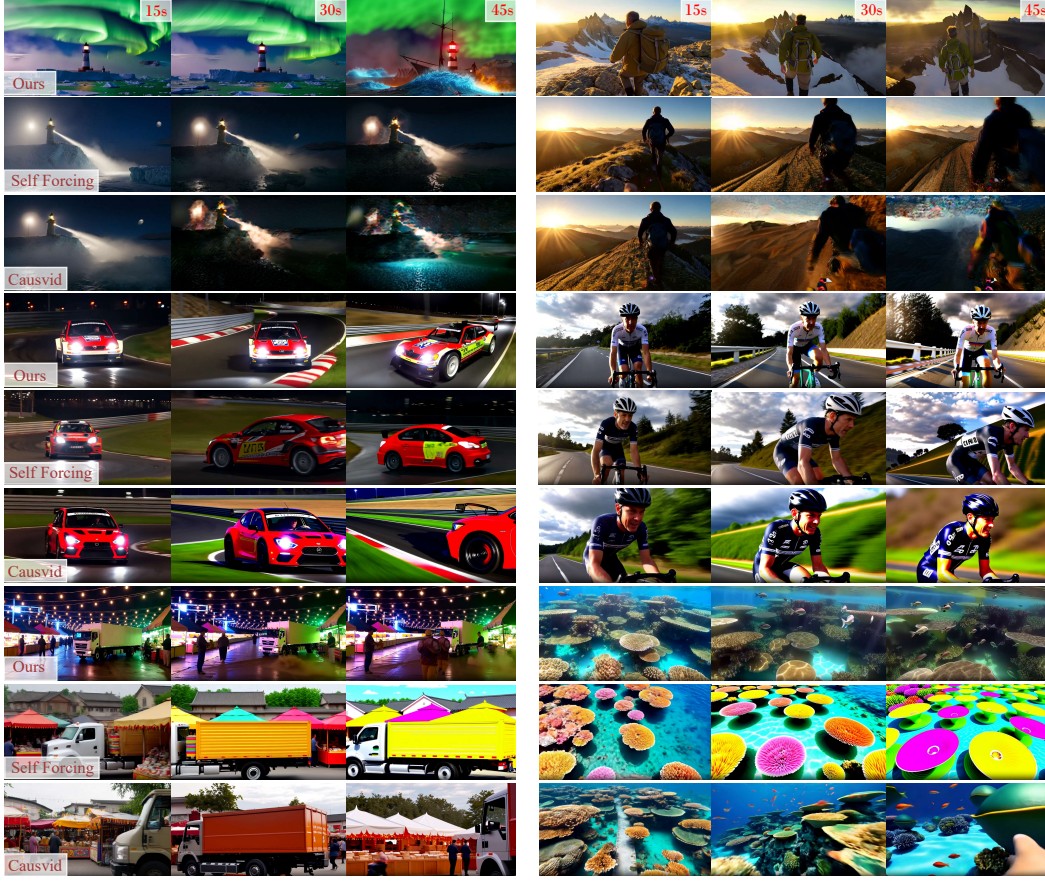

Figure 7: Qualitative comparison of long video generation (45s) with Self-Forcing and Causvid. The visual results show that other methods suffer from noticeable saturation distortion and quality decay over time, whereas our approach preserves detail and consistency. Additional results are provided in the supplementary material.

### 4.3 ABLATION STUDIES

**Key Components.** Diagonal Denoising assigns more denoising steps to early video chunks to establish a high-quality foundation and progressively reduces denoising steps for subsequent chunks, whereas without it, the same number of steps is applied uniformly across all chunks. Diagonal Forcing refers to using noisy frames instead of clean frames as the Key-Value (KV) cache in autoregressive generation. Our ablation study shows that removing either flow distribution matching loss or Diagonal Forcing significantly degrades video quality across all metrics (Table 2). Without Diagonal Denoising (this corresponds to the inference cost of Self-Forcing in Table 1), we observe that the model achieves performance comparable to ours, while our method attains a $1.53\times$ speedup. Notably, we find that the flow distribution matching loss primarily benefits the few-step denoising

| Ablation Variant | Temporal Quality ↑ | Frame Quality ↑ | Text Alignment ↑ | Total Score ↑ |
|---|---|---|---|---|
| Without Diagonal Forcing | 92.1 | 60.1 | 26.9 | 83.58 |
| Without Flow Loss | 92.5 | 60.8 | 27.8 | 84.18 |
| Without Diagonal Denoising | **95.1** | 63.2 | 28.6 | 84.46 |
| **Full Method (Ours)** | 94.9 | **63.4** | **28.9** | **84.48** |

Table 2: Ablation Study on Key Components of DiagDistill.

| Steps | Temporal Quality ↑ | Frame Quality ↑ | Text Alignment ↑ | NFEs | In-Flight Latency (s) ↓ | Throughput (FPS) ↑ |
|---|---|---|---|---|---|---|
| 4322222 | 94.9 | 63.4 | 28.9 | 34 | 0.23 ± 0.02 | 31.0 |
| 5433333 | 95.1 | 63.2 | 29.3 | 48 | 0.34 ± 0.02 | 23.3 |
| 5432222 | 94.8 | 63.1 | 29.0 | 40 | 0.23 ± 0.02 | 29.7 |
| 5333333 | 95.0 | 63.9 | 29.1 | 46 | 0.34 ± 0.02 | 22.5 |
| 4333333 | 95.0 | 63.7 | 28.5 | 44 | 0.34 ± 0.02 | 23.5 |
| 4222222 | 93.4 | 62.3 | 27.8 | 32 | 0.23 ± 0.02 | 32.0 |

Table 3: Evaluation of denoising step configurations.

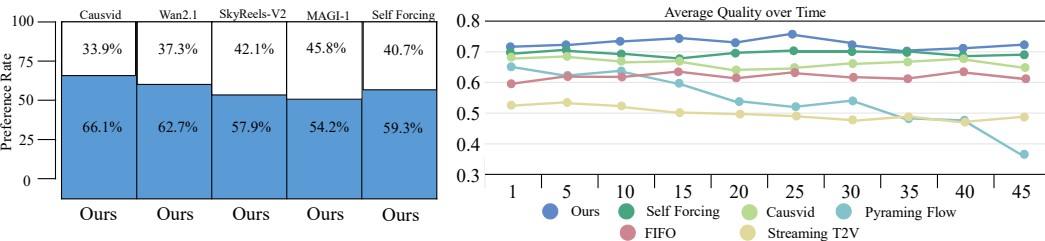

Figure 8: Quantitative evaluation of long video generation. The plot compares human preference scores and quality consistency over time for different methods under identical conditions. The results show that our approach maintains a stable quality throughout extended sequences, achieving scores above 50%, and attains a significant reduction in inference latency.

regime by aligning its performance with the many-step denoising baseline (*i.e.*, without Diagonal Denoising), while offering limited gains when applied in a many-step denoising setting.

**Diagonal Forcing Timesteps.** Moreover, we systematically evaluated diagonal forcing using metrics across different noise levels of timesteps for the kv cache. As Figure 5(a) shows, 100 timesteps achieved optimal scores across all evaluation dimensions, including temporal quality, frame quality, and text alignment. The performance peaks at this specific noise level before degrading as timesteps approach complete noise addition (1000 steps) or clean frames (0 steps). This can be attributed to the fact that excessive noise (high timesteps) blurs the structural priors in the video context, leading to reduced motion magnitude. This also explains why our method generates larger motion amplitudes compared to MAGI (Teng et al., 2025). Conversely, insufficient noise (low timesteps) causes the next chunk prediction to implicitly perform next noise level prediction, which can result in over-denoising of subsequent chunks and ultimately lead to over-saturated outputs.

**Flow Loss Weight.** We conducted a comprehensive ablation study across eight motion loss weight configurations. Figure 5(b) reveals the crucial balance between motion guidance (via Flow Distribution Matching) and the DMD learning objectives, with optimal performance observed at a weight of 1.0. This balanced weighting scheme ensures the harmonious optimization of temporal consistency, frame quality, and textual alignment metrics.

**Denoising Configurations.** We evaluated six denoising configurations (represented by 7-digit sequences specifying steps per chunk as a 5 seconds video have 7 chunks in our setting) across quality and computational metrics. As shown in Table 3, these configurations exhibit trade-offs between generation quality and efficiency. Among them, configuration 5333333 achieves the highest quality, while 4222222 offers the maximum throughput. To balance video quality and real-time performance, we selected configuration 4322222, as it has the second-lowest number of NFEs and delivers performance comparable to configurations with significantly higher latency and throughput, with only marginal differences.

### 4.4 LONG VIDEO GENERATION EVALUATION

We evaluated our long video generation framework using both simple and complex prompts. As shown in Figure 8, our model maintains consistent perceptual quality over time, whereas baseline methods suffer from rapid quality decay due to error accumulation. A large-scale user study (93 participants, 150 comparisons per model pair) on the first 50 prompts from MovieGenBench further validated our method's superiority in overall visual quality, text faithfulness, and long-term consistency User study results, consistent with the qualitative comparison in Figure 7, confirm that baseline methods suffer from degradation issues such as saturation distortion, whereas our approach consistently maintains high visual quality. A key feature of our framework is its support for *dynamic*

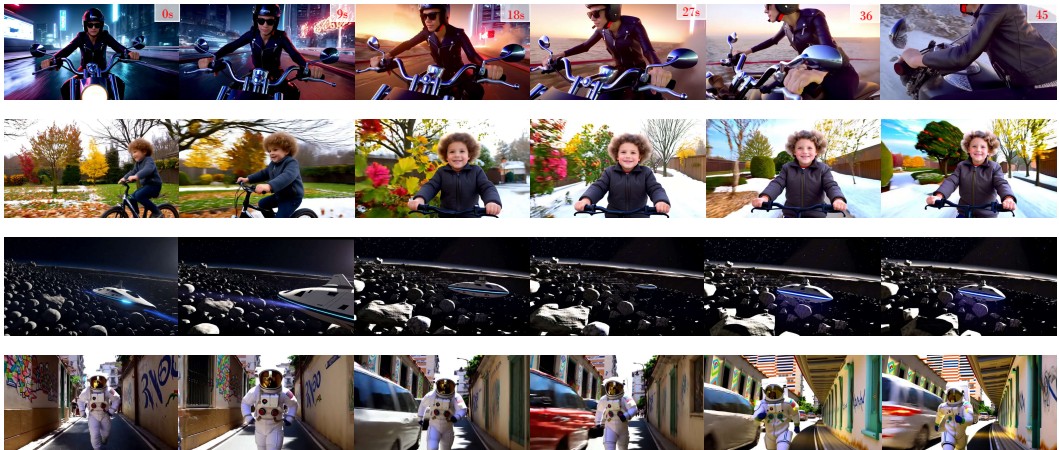

Figure 9: Illustration of long video generation with dynamic prompting. This feature allows for the integration of new prompts at arbitrary time points, facilitating the creation of coherent long videos with changing narratives. The specific prompts used for each segment are detailed in the appendix.

*prompting* (Figure 9), which enables users to introduce new text descriptions at any point along the timeline to create complex narratives with evolving scenes and actions.

## 5    CONCLUDING REMARKS

In this work, we present Diagonal Distillation, a novel framework for efficient autoregressive video generation that explicitly accounts for the temporal structure of the denoising process. Our approach exploits dependencies across both video chunks and denoising steps through an asymmetric denoising strategy, in which more steps are allocated to early chunks and progressively fewer to later ones. This design reflects the observation that early chunks play a more critical role in establishing global motion and appearance, allowing us to substantially reduce the overall number of denoising steps without degrading motion coherence or visual fidelity.

To further improve temporal stability, Diagonal Forcing explicitly models the denoising trajectory along the temporal dimension, mitigating error accumulation across chunks and narrowing the mismatch between training and inference dynamics. This alignment leads to more stable long-range synthesis and reduces drift in extended video generation. In addition, Flow Distribution Matching enforces dynamic consistency under strict step constraints by aligning the optical flow distributions of generated and real videos, thereby preserving realistic motion patterns even in low-step regimes. Extensive experiments demonstrate that our method consistently achieves a superior trade-off between computational efficiency and generation quality, enabling scalable long-horizon video synthesis while maintaining strong temporal coherence.

## ACKNOWLEDGEMENT

The authors would like to thank all the anonymous reviewers for providing helpful suggestions to improve this paper. Authors with equal contributions are listed in alphabetical order and allowed to change their orders freely on their resume and website.

## ETHICS STATEMENT

While the real-time video generation technology presented in this study significantly improves generation efficiency (achieving a 277.3× speedup compared to the baseline model), we are fully aware of its dual-use nature. This technology could potentially be misused to create misleading content or deepfake videos. To mitigate this risk, we commit to embedding usage guidelines and restrictions when open-sourcing the code and models, and we advocate for the adoption of traceability technologies such as digital watermarks and content authentication. Concurrently, this technology holds significant positive potential in fields such as education, the creative industries, and assistive tools. We aim to maximize its societal benefits and minimize potential harms through ongoing discussions on technology ethics and responsible release practices.

## REPRODUCIBILITY STATEMENT

For detailed reproducibility information, including full implementation details, training configurations, hyperparameters, and evaluation protocols, please refer to the appendix sections. All source code, trained model weights, and configuration files will be released to ensure the full reproducibility of our results.

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

# Appendix

## Table of Contents

## A    USE OF LLMS

In this study, Large Language Models (LLMs) were used exclusively for the purpose of checking and correcting grammatical errors in the manuscript.

## B    NOISE SCHEDULING AND MODEL PARAMETERIZATION

Following the design principles of the Wan 2.1 series, we adopt a Flow Matching framework, utilizing a time step offset defined as:

$$t'(k, t) = \frac{(k \cdot t/1000)}{1 + (k-1)(t/1000)} \times 1000$$

with an offset factor $k = 5.0$. The forward process is defined as:

$$\mathbf{x}_t = \frac{t'}{1000} \times \mathbf{x} + \left(1 - \frac{t'}{1000}\right) \times \boldsymbol{\varepsilon}, \quad \boldsymbol{\varepsilon} \sim \mathcal{N}(\mathbf{0}, \mathbf{I})$$

where the time step $t \in [0, 1000]$. The data prediction model is expressed as:

$$G_{\boldsymbol{\theta}}(\mathbf{x}, t, c) = c_{\text{skip}} \cdot \boldsymbol{\varepsilon} - c_{\text{out}} \cdot \mathbf{v}_{\boldsymbol{\theta}}(c_{\text{in}} \cdot \mathbf{x}_t, c_{\text{noise}}(t'), c)$$

We maintain the preconditioning coefficients identical to the base model configuration, i.e., $c_{\text{skip}} = c_{\text{in}} = c_{\text{out}} = 1$ and $c_{\text{noise}}(t) = t$.

Based on our actual configuration, DMD (Distribution Matching Distillation) training employs a Diagonal Denoising mode with a time step list of [1000, 100]. The enabling of time step wrapping is controlled by the `warp_denoising_step` parameter.

## C    COMPARISON OF TEMPORAL TRAINING STRATEGIES AND MODEL ARCHITECTURE DETAILS

This appendix explains Figure 10, which compares temporal training strategies for autoregressive video generation, highlighting our **Diagonal Forcing** approach. The figure shows how a Causal DiT model generates new frames conditioned on previous ones, with strategy differences affecting robustness and performance.

The standard baseline approaches include Teacher Forcing, Diffusion Forcing, and Self Forcing training strategies, as shown in Figure 10. Teacher Forcing conditions the model exclusively on ground-truth previous frames during training, providing a clean learning signal but creating a train-inference discrepancy that leads to error compounding during long sequences. Diffusion Forcing addresses this by exposing the model to noisy latents from the diffusion process, enhancing robustness but introducing a mismatch since the model learns to denoise based on noisy context unlike the clean context used at inference. Self Forcing directly mimics inference by conditioning on the model's own previous predictions, but this approach trains the model on its own early, often low-quality predictions, which can hinder learning and lead to suboptimal convergence.

Our proposed Diagonal Forcing method, illustrated in Figure 10(d), combines the advantages of these strategies while mitigating their weaknesses. The core innovation conditions the denoising of the current frame on a mixture of clean ground-truth and previously generated frames, arranged in a diagonal pattern across the temporal dimension. For a given target frame, the model is conditioned on previous frames where the most recent ones are its own predictions from previous autoregressive steps, while frames further in the past are drawn from ground-truth data. This pattern shifts diagonally for each subsequent target frame, providing two crucial benefits: it gradually exposes the model to its own prediction errors to improve long-term generation robustness, while simultaneously anchoring the process on distant, clean ground-truth frames to prevent semantic drift. Most importantly, this conditioning pattern closely approximates the inference-time scenario of our Diagonal Denoising algorithm, directly aligning training and inference distributions for generating long, coherent videos.

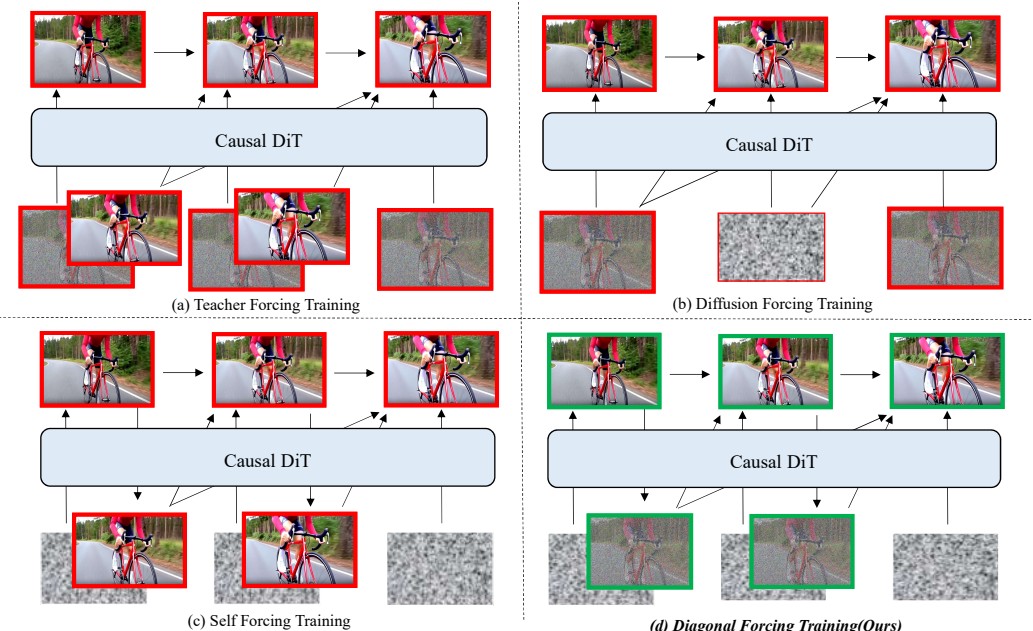

Figure 10: Comparative visualization of temporal training strategies for autoregressive video generation using Causal DiT. Four panels illustrate: (a) Teacher Forcing (green boxes for ground-truth frames), (b) Diffusion Forcing (red boxes for noisy latents), (c) Self Forcing (red boxes for model's own predictions), and (d) **Diagonal Forcing (Ours)** (mixed green/red boxes in diagonal patterns). Each row represents sequential frame generation, with arrows indicating causal dependencies. The diagonal pattern in (d) highlights the core innovation—blending clean past frames with recent model-generated ones to align training/inference distributions. This visual comparison underscores how Diagonal Forcing bridges gaps in robustness and coherence seen in baseline methods.

## D  PSEUDO-CODE FOR DIAGONAL DENOISING WITH NOISY KV CACHE

This appendix provides the pseudo-code for Algorithm 1, which implements the Diagonal Denoising with Noisy KV Cache strategy introduced in Section 3.2 and Figure 3. The algorithm 1 enables efficient long video generation through progressive step reduction and KV cache reuse. The algorithm processes video chunks sequentially. During the **Base Phase** ($k \leq 4$), each chunk is denoised for $\mathbf{s}[k]$ steps (progressively reducing from 5 to 2). At the penultimate step, the intermediate latent is processed by CACHENOISYRESULT, which adds noise to create $\tilde{X}^{\text{interim}}$ and caches its KV embeddings. This noisy caching simulates the teacher's denoising trajectory, providing temporal context for subsequent chunks.

In the **Extension Phase** ($k > 4$), generation uses only 2 steps. Chunk $k$ is conditioned on the previous chunk's output ($\tilde{X}_{k-1}$), and the first step's result is similarly cached. This approach maintains coherence while significantly improving efficiency, balancing quality and computational cost for long video generation.

## E  DETAILED ANALYSIS OF ACCELERATION AND STEP ALLOCATION

This section provides a comprehensive analysis of the acceleration mechanisms in Diagonal Forcing and explores various denoising step allocation strategies.

### E.1  ACCELERATION ANALYSIS

The acceleration in Diagonal Forcing stems from four synergistic optimizations that collectively achieve superior performance compared to Self-Forcing:

**1. Reduction in Denoising Steps:** Diagonal Forcing achieves comparable or better quality with fewer total Noise Function Evaluations (NFEs). While our method naturally reduces the number

---

**Algorithm 1** Diagonal Denoising with Noisy KV Cache

---

**Require:** $M$, $\mathbf{s} = [5, 4, 3, 2, 2, 2, 2]$, $\{\Theta_s, \Theta_2\}$
**Ensure:** $\tilde{X}_1, \ldots, \tilde{X}_M$
 1: $\tilde{X} \leftarrow \emptyset, \mathcal{C} \leftarrow \emptyset$
 2: **for** $k = 1$ **to** $M$ **do**
 3:     $\epsilon \sim \mathcal{N}(0, I)$
 4:     **if** $k \leq 4$ **then**                                           ▷ Base phase
 5:         $X_k^{(0)} \sim \mathcal{N}(0, I)$
 6:         **for** $t = 1$ **to** $\mathbf{s}[k]$ **do**
 7:             $X_k^{(t)} \leftarrow D^{(1)}(X_k^{(t-1)}, \tilde{X}_{<k}, \mathcal{C}; \Theta_{\mathbf{s}[k]})$
 8:             **if** $t = \mathbf{s}[k] - 1$ **then**
 9:                 CACHENOISYRESULT$(X_k^{(t)})$                     ▷ Cache at penultimate step
10:             **end if**
11:         **end for**
12:         $\tilde{X}_k \leftarrow \text{Mix}(X_k^{(\mathbf{s}[k])}, \epsilon)$
13:     **else**                                        ▷ Extension phase (2-step generation)
14:         $C_k \leftarrow \mathcal{T}(\tilde{X}_{k-1})$
15:         $X_k^{\text{step1}} \leftarrow D^{(1)}(\mathcal{N}(0, I), C_k, \mathcal{C}; \Theta_2)$
16:         CACHENOISYRESULT$(X_k^{\text{step1}})$
17:         $\tilde{X}_k \leftarrow \text{Mix}(D^{(2)}(X_k^{\text{step1}}, C_k, \mathcal{C}; \Theta_2), \epsilon)$
18:     **end if**
19:     $\tilde{X} \leftarrow \tilde{X} \cup \{\tilde{X}_k\}$
20: **end for**
21: **return** $\tilde{X}$
22: **procedure** CACHENOISYRESULT$(X)$
23:     $\tilde{X}^{\text{interim}} \leftarrow \sqrt{\alpha_k} X + \sqrt{1 - \alpha_k} \epsilon$
24:     $\mathcal{C} \leftarrow \mathcal{C} \cup \{\text{ComputeKV}(\tilde{X}^{\text{interim}})\}$
25: **end procedure**

---

of required denoising steps, we conducted controlled experiments under identical NFE budgets to isolate the contributions of other optimizations. As shown in Table 5, even with the same number of NFEs, Diagonal Forcing consistently outperforms Self-Forcing across all quality metrics while delivering lower latency and higher throughput.

**2. Efficient KV Cache Mechanism:** Another architectural advantage lies in our integrated KV cache strategy. Self-Forcing requires separate KV cache computations on clean frames without performing denoising, creating redundant operations. In contrast, our method performs KV caching directly on the noisy latent as shown in Figure 10 and apply tiny vae as our tokenization (Boer Bohan, 2025), which simultaneously serves two purposes: providing conditioning for subsequent chunks and progressing the denoising process to yield clean latents. This elimination of redundant computations directly translates to reduced both denoising and decoding latency and increased throughput.

**3. Optimized Attention Window Size:** Our rolling KV cache mechanism, inspired by Self-Forcing, employs a more efficient context window strategy. While maintaining seamless rolling forward to avoid sliding-window overhead, we reduce the KV cache size from 6 chunks (used in original Self-Forcing) to 4 chunks. This optimization is supported by our scaling analysis in Table 4, which shows that performance plateaus around window sizes of 12-27, with smaller windows providing better latency-memory trade-offs. The reduced window size contributes significantly to lower memory usage and faster inference and lower in-flight latency without compromising long-video generation quality.

**4. Efficient Tokenization Mechanism:** The tokenization process operates directly on the latent representations from the Tiny VAE (Boer Bohan, 2025), which encodes frames into tokens. As shown in Table 8, the Tiny VAE achieves substantially faster decoding with significantly fewer parameters than the Full VAE. For streaming applications, we further optimize the pipeline using an efficient Tiny VAE. By eliminating the VAE bottleneck, the Tiny VAE reduces decoding time by more than 10×.

The combination of these four optimizations creates a compound effect: the reduced denoising steps decrease computational load, the efficient KV cache mechanism eliminates redundant operations,

| Attention Window Size | Total Score | In-Flight Latency (s) | Memory (GB) |
|---|---|---|---|
| 3 | 80.9 | $0.37 \pm 0.01$ | 14.9 |
| 6 | 81.3 | $0.38 \pm 0.01$ | 15.8 |
| 9 | 82.3 | $0.40 \pm 0.01$ | 16.6 |
| 12 | 84.3 | $0.46 \pm 0.02$ | 17.5 |
| 15 | 84.2 | $0.51 \pm 0.02$ | 18.4 |
| 18 | 84.4 | $0.54 \pm 0.02$ | 19.2 |
| 21 | 84.5 | $0.59 \pm 0.02$ | 20.1 |
| 24 | 84.3 | $0.64 \pm 0.02$ | 20.9 |
| 27 | 84.5 | $0.68 \pm 0.02$ | 21.8 |

Table 4: KV cache scaling analysis

| Total NFEs | Mode | Steps Allocation | Temporal Quality ↑ | Frame Quality ↑ | Text Alignment ↑ | In-Flight Latency (s) ↓ | Throughput (FPS) ↑ |
|---|---|---|---|---|---|---|---|
| 34 | Diagonal | 4322222 | 94.9 | 63.4 | 28.9 | $0.23 \pm 0.02$ | 31.0 |
| 34 | Self | 4322222 | 93.5 | 62.1 | 27.5 | $0.43 \pm 0.02$ | 25.9 |
| 40 | Diagonal | 5432222 | 94.8 | 63.1 | 29.0 | $0.23 \pm 0.02$ | 29.7 |
| 40 | Self | 5432222 | 93.9 | 62.3 | 28.1 | $0.43 \pm 0.02$ | 23.8 |
| 44 | Diagonal | 4333333 | 95.0 | 63.7 | 28.5 | $0.34 \pm 0.02$ | 23.5 |
| 44 | Self | 4333333 | 94.2 | 62.8 | 27.8 | $0.56 \pm 0.02$ | 21.5 |
| 46 | Diagonal | 5333333 | 95.0 | 63.9 | 29.1 | $0.34 \pm 0.02$ | 22.5 |
| 46 | Self | 5333333 | 94.5 | 63.0 | 28.6 | $0.56 \pm 0.02$ | 19.8 |
| 48 | Diagonal | 5433333 | 95.1 | 63.2 | 29.3 | $0.34 \pm 0.02$ | 23.3 |
| 48 | Self | 5433333 | 94.3 | 62.5 | 28.4 | $0.56 \pm 0.02$ | 18.6 |
| 56 | Diagonal | 4444444 | 95.1 | 63.2 | 28.6 | $0.46 \pm 0.02$ | 21.5 |
| 56 | Self | 4444444 | 94.3 | 63.1 | 29.3 | $0.69 \pm 0.02$ | 17.0 |

Table 5: Comparative evaluation under identical NFE budgets demonstrates Diagonal Forcing's superior efficiency. Our method achieves better quality metrics with significantly lower latency and higher throughput across all configurations.

the optimized window size minimizes memory overhead and the tiny vae reduce the extra decoding time. This holistic approach enables Diagonal Forcing to achieve the performance advantages demonstrated in our comparative evaluations, making it particularly suitable for long-video generation scenarios where both quality and efficiency are critical.

## E.2 STEP ALLOCATION STRATEGY ANALYSIS

| Schedule | Temporal Quality ↑ | Frame Quality ↑ | Text Alignment ↑ | NFEs | In-Flight Latency (s) ↓ | Throughput (FPS) ↑ |
|---|---|---|---|---|---|---|
| *Baseline Strategies (From Paper)* | | | | | | |
| 5333333 | 95.0 | 63.9 | 29.1 | 46 | $0.34 \pm 0.02$ | 22.5 |
| 4333333 | 95.0 | 63.7 | 28.5 | 44 | $0.34 \pm 0.02$ | 23.5 |
| 5433333 | 95.1 | 63.2 | 29.3 | 48 | $0.34 \pm 0.02$ | 23.3 |
| 5432222 | 94.8 | 63.1 | 29.0 | 40 | $0.23 \pm 0.02$ | 29.7 |
| 4322222 | 94.9 | 63.4 | 28.9 | 34 | $0.23 \pm 0.02$ | 31.0 |
| 4222222 | 93.4 | 62.3 | 27.8 | 32 | $0.23 \pm 0.02$ | 32.0 |
| *Exploration: Emphasizing Early Frames* | | | | | | |
| 5422222 | **95.1** | **64.1** | 29.2 | 38 | $0.23 \pm 0.02$ | 28.5 |
| 5522222 | 95.3 | 63.8 | **29.4** | 40 | $0.23 \pm 0.02$ | 27.1 |
| 4432222 | 94.9 | 63.5 | 28.8 | 38 | $0.23 \pm 0.02$ | 30.2 |
| *Exploration: Non-monotonic / Dynamic Allocation* | | | | | | |
| 4343232 | 95.0 | 63.6 | 28.4 | 42 | $0.34 \pm 0.02^*$ | 21.0 |
| 4532232 | 95.0 | 63.7 | 28.6 | 42 | $0.34 \pm 0.02^*$ | 20.5 |
| 3333533 | 93.5 | 60.0 | 27.7 | 46 | $0.37 \pm 0.02^*$ | 22.8 |
| 3233343 | 93.6 | 60.9 | 27.9 | 42 | $0.34 \pm 0.02^*$ | 23.3 |

Table 6: Quantitative comparison of different denoising step allocation strategies. Strategies are grouped by type: Baseline Strategies (evaluated in the main paper), Exploration: Emphasizing Early Frames, and Exploration: Non-monotonic / Dynamic Allocation. NFEs denotes the total number of function evaluations. Best results are in bold.

To thoroughly investigate the impact of denoising step allocation on the performance of our distilled video diffusion model, we extend the analysis beyond the baseline strategies presented in the main paper. We systematically explore a wider range of allocation schedules, which can be broadly categorized into two groups: strategies that emphasize early frames and those employing non-monotonic

or dynamic allocations. The quantitative comparison of these strategies is summarized in Table 6. For non-monotonically decreasing allocations, we report the average latency, which is provided for reference only and is marked with an asterisk (*).

- **Performance Upper Bound:** We confirm that using 4 steps for the initial chunk (400000) essentially reaches the performance upper bound for our distillation framework. Strategies like 5422222 show marginally better metrics than 4322222, but the improvement is minimal and comes with latency cost. This suggests that for our DMD framework, using 4 steps for the initial generation is often sufficient to approach the performance upper bound. Further increasing steps (e.g., to 5) provides diminishing returns.

- **Monotonic Decrease is Optimal:** The exploration of non-monotonic schedules (e.g., 4343232) reveals that they do not provide a consistent or significant advantage. More importantly, they can harm motion quality. In our autoregressive framework, subsequent chunks are generated conditioned on the motion characteristics of prior chunks. If the initial chunks have strong motion (established by sufficient denoising steps), later chunks can maintain good motion even with fewer steps. Introducing more steps later does not effectively enhance motion and can disrupt this inherent consistency. Therefore, a monotonically decreasing schedule is the most sensible approach.

- **Limited Search Space:** The solution space for effective schedules is relatively small. The principle of "more steps early, fewer steps later" is robust. For practical application, manually selecting a schedule like 4322222 (for 5s) and cyclically extending it for longer videos (43222224322222 for 10s) is a robust, effective, and recommended practice. Users can adjust the initial step count (3, 4, or 5) based on their specific latency-quality requirements.

In conclusion, while algorithmic schedule selection confirms the robustness of our manually tuned strategy, it ultimately reinforces that a simple, monotonically decreasing allocation cyclically extended for longer videos is the most effective and principled rule.

## F    TECHNICAL DETAILS OF THE STREAM PROTOCOL

In this section, we provide our streaming protocol is detailed as follows:

**(1) Chunk Size**    The core component is a rolling KV cache mechanism following the Self-Forcing approach (Huang et al., 2025), operating with a chunk size of 3 frames. This chunk size was empirically selected based on our teacher model's VAE latent space and provides the optimal trade-off. Comparative analysis in Table 7 shows that the chunk-level variant (size=3) achieves a total score of 84.48, while the frame-level variant (size=1) scores 84.29. The chunk-level variant provides higher throughput (31 FPS vs. 16.5 FPS) at the cost of slightly higher latency (0.37s vs. 0.25s). This observation aligns with Self-Forcing [1], where frame-level configurations are more suitable for real-time applications on constrained devices but result in inferior video modeling performance.

| Type | Throughput (FPS) ↑ | Latency (s) ↓ | Speedup ↑ | Total Score | Quality | Semantics |
|---|---|---|---|---|---|---|
| DiagDistill (Chunk-level, chunk size=3) | 31.0 | 0.37 | 277.3x | 84.48 | 85.26 | 81.73 |
| DiagDistill (Frame-level, chunk size=1) | 16.5 | 0.25 | 412.0x | 84.29 | 85.41 | 80.40 |

Table 7: Performance comparison of chunk-level (size=3) vs. frame-level (size=1) streaming variants

**(2) Overlap, Buffering, and KV Cache Size**    Our buffering strategy employs a fixed-size KV cache that maintains context from the most recent 4 chunks, resulting in a consistent memory footprint of 17.5GB. This represents a significant improvement over the original Self-Forcing approach, which requires 19.2GB (with 18-frame KV cache reuse equivalent to 6 chunks overlap). The latent space overlap is 9 frames. We use the denoising state immediately preceding the final clean step as conditioning for the KV cache. This approach completes the denoising process while preserving the noise-conditioned frames for injection into the KV cache sequence. Each chunk is generated using 4 denoising steps, with total scores evaluated on VBench-Long. As shown in Table 4 from our paper, the KV cache size scaling analysis reveals critical performance trade-offs associated with attention window size. As the window increases from 3 to 27 frames, the total score improves significantly from 80.9 to 84.5, but diminishing returns are observed beyond 12 frames. This performance improvement comes at a cost: latency increases linearly from 0.37s to 0.68s due to the larger context

that must be processed, and memory footprint grows substantially from 14.9GB to 21.8GB. The optimal operating point is therefore determined to be 12 frames–our chosen configuration–which effectively balances a high total score of 84.3, reasonable latency of 0.46s, and manageable memory usage of 17.5GB.

**(3) Tokenization**    The tokenization process operates directly on the tiny VAE latent space representations (Boer Bohan, 2025), where frames are encoded as tokens. As shown in Table 8, Tiny VAE achieves substantially faster decoding with significantly fewer parameters compared to the Full VAE. For streaming demos, we further optimize the pipeline with an efficient Tiny VAE. Tiny VAE removes the VAE bottleneck by reducing the decoding time over 10×.

| Model | Decoder Params | Compression Rate | Decoding Time(s) $(81 \times 832 \times 480)$ |
|---|---|---|---|
| Full VAE (Wan 2.1) | 73.3M | $8 \times 8 \times 4$ | 1.67 |
| Tiny VAE (Wan 2.1) (Boer Bohan, 2025) | 9.84M | $8 \times 8 \times 4$ | 0.12 |

Table 8: Comparison of Causal VAE Models.

**(4) Pre-fill Cost**    The reported "first-frame latency" includes the complete multi-step denoising process for the initial chunk. For all subsequent chunks, the pre-fill cost is effectively eliminated as the model leverages the rolling cache, with computation focused solely on denoising new frames. This architecture, with its favorable O(TL) time complexity, has been extensively validated on long sequences (exceeding 45 seconds) with no measurable performance degradation, confirming its robustness for real-time deployment.

## G    Detailed Ablation Study on Motion Flow Field Representation

Motivated by MCM (Zhai et al., 2024) and as detailed in our paper, our method eschews reliance on external, pre-trained optical flow estimators (e.g., RAFT (Teed & Deng, 2020)). Instead, we introduce a lightweight, learnable motion feature extraction module, $\mathcal{F}(\cdot)$, that operates directly on the latent space of the video diffusion model. A key component of our implementation is an Exponential Moving Average (EMA) framework. Within this framework, the student component of $\mathcal{F}(\cdot)$ is trained with gradient backpropagation, enabling end-to-end motion representation learning. The teacher component is updated via EMA of the student's parameters. We apply standard feature normalization to the latent features, and this self-supervised distillation framework inherently avoids biases that could be introduced by errors from external flow estimators. We investigated several design variants for the motion feature extractor $\mathcal{F}(\cdot)$:

1. **Latent Difference:** Computes frame-wise differences between consecutive latent representations: $F_{\text{diff}}(X) = X_{t+1} - X_t$

2. **Latent Correlation:** Constructs 4D correlation volumes via pairwise dot-products: $F_{\text{corr}}(X) = \text{Corr4D}(X_t, X_{t+1})$ (Shi et al., 2023)

3. **Latent Frequency Components:** Extracts low-frequency components ($F_{\text{low}}$) and high-frequency components ($F_{\text{high}}$) using DCT transformations (Wu et al., 2024).

4. **Learnable Representation (MLP):** Applies a two-layer MLP to adaptively extract motion features. This MLP contains no temporal layers, thus preserving motion information while avoiding direct application of appearance loss to the latent representation. The target MLP head is updated via exponential moving average (EMA) and is discarded during inference. This design is analogous to the projection heads used in self-supervised learning (Chen et al., 2020).

5. **Learnable Representation with Convolution on Latent Difference:** First computes latent differences between consecutive frames, then applies convolutional layers to extract local motion patterns. This hierarchical design captures both explicit motion cues from frame differences and local spatiotemporal correlations through convolutional processing, effectively implementing the motion flow field extraction function $F(x)$ described in our flow distribution matching framework.

For the learnable representations (variants 4 and 5), we employ an EMA-based framework to ensure training stability:

- **Student Network:** Parameters $\theta$ are updated via gradient descent.
- **Target Network:** Parameters $\theta^-$ are updated via EMA: $\theta^- \leftarrow \mu \cdot \theta^- + (1 - \mu) \cdot \theta$.

The teacher's output is processed by the target MLP and the student's output by the student MLP. EMA ensures stable target evolution, improved feature quality, and prevents model collapse.

The computational overhead of our flow module is minimal. It consists of a very light Conv-MLP with only two convolutional layers, making its computational cost almost negligible. This is especially true when compared to the alternative approach of decoding frames with a VAE and then applying a separate optical flow estimator, which would incur significant memory overhead.

The regression loss term is critical for stable optimization. Without it, training is highly prone to collapse. The regression loss provides a crucial grounding signal that constrains the optimization trajectory. This is corroborated by our ablation study on the flow loss weight (see Figure 5b), which shows that performance degrades if the weight is set excessively high. This indicates that an overly strong flow constraint is detrimental. A comprehensive ablation over eight configurations confirmed that a weight of 1.0 achieves the optimal balance, ensuring harmonious optimization of temporal consistency, frame quality, and textual alignment.

| Flow Representation Variant | Temporal Quality ↑ | Frame Quality ↑ | Text Alignment ↑ |
|---|---|---|---|
| Raw Latent | 92.5 | 60.8 | 27.8 |
| Latent Difference | 93.8 | 61.5 | 28.2 |
| Latent Correlation | 94.2 | 62.1 | 28.4 |
| Low Frequency Components | 93.5 | 61.8 | 28.1 |
| High Frequency Components | 94.0 | 62.3 | 28.5 |
| Learnable MLP | 94.6 | 63.2 | 28.7 |
| Learnable + Conv on Diff | **94.9** | **63.4** | **28.9** |

Table 9: Performance comparison of different $\mathcal{F}(\cdot)$ implementations.

# H  USER STUDY

## H.1  EXPERIMENTAL SETUP

To comprehensively evaluate the quality and consistency of long video generation methods, we conducted a large-scale double-blind user study involving 93 participants. Each participant completed 150 comparisons per model pair, resulting in statistically significant findings. The evaluation was performed on the first 50 prompts from MovieGenBench to ensure comprehensive assessment across diverse scenarios.

## H.2  EVALUATION METHODOLOGY

### EVALUATION CRITERIA

Participants compared different methods based on three key aspects:

- **Overall Visual Quality:** Subjective assessment of visual fidelity, naturalness, and absence of visual artifacts
- **Text Faithfulness:** Accuracy in matching the input text description and maintaining semantic alignment
- **Long-term Consistency:** Quality maintenance across extended sequences (up to 45 seconds)

### COMPARATIVE FRAMEWORK

The study employed a paired comparison design where participants evaluated:

- Side-by-side comparisons of our method against five baseline approaches
- Videos generated from identical prompts under controlled conditions
- Both simple and complex prompt scenarios to test generalization capability
- Dynamic prompting scenarios with multiple scene transitions

### H.3 USER STUDY RESULTS

The quantitative results from the user study demonstrate clear superiority of our method, as show in our paper. The preference rates are summarized as follows:

- **vs. Causvid:** 66.1% preference for our method (33.9% for baseline)
- **vs. WAN2.1:** 62.7% preference for our method (37.3% for baseline)
- **vs. SkyReels-V2:** 57.9% preference for our method (42.1% for baseline)
- **vs. MAGI-1:** 54.2% preference for our method (45.8% for baseline)
- **vs. Self-Forcing:** 59.3% preference for our method (40.7% for baseline)

Statistical analysis using paired t-tests confirmed that all preference rates are statistically significant ($p < 0.01$), indicating robust user preference for our approach across all baseline comparisons.

### H.4 QUESTIONNAIRE DESIGN AND IMPLEMENTATION

#### PARTICIPANT DEMOGRAPHICS

The study involved 93 participants with diverse backgrounds:

- **Background distribution:** Computer Science (42%), AI/ML (28%), Visual Arts (18%), Other (12%)
- **Experience levels:** Beginner (15%), Intermediate (45%), Expert (40%)
- **Familiarity with video generation:** Average rating of 3.8/5.0

#### QUESTIONNAIRE STRUCTURE

1. **Demographic Information Collection**
   - Professional background and expertise level assessment
   - Prior experience with video generation technologies

2. **Individual Video Quality Assessment**
   - 5-point Likert scale evaluation (1=Poor, 5=Excellent)
   - Assessment of visual realism, color consistency, motion naturalness
   - Artifact detection and severity rating

3. **Comparative Evaluation Tasks**
   - Side-by-side method comparisons with forced-choice preference
   - Specific aspect comparisons (text alignment, temporal consistency)
   - Quality assessment at multiple time points (15s, 30s, 45s)

4. **Dynamic Prompting Evaluation**
   - Transition smoothness assessment for multi-scene narratives
   - Narrative coherence evaluation across prompt changes
   - Visual consistency maintenance during scene transitions

5. **Qualitative Feedback Collection**
   - Open-ended comments on observed artifacts
   - Identification of temporal consistency issues
   - Suggestions for methodological improvements

### H.5 DETAILED QUESTIONNAIRE EXAMPLE

**Part 1: Demographic Information**

| Item | Options |
|---|---|
| Background | ☐ Computer Science ☐ AI/ML ☐ Visual Arts ☐ Other: ________ |
| Experience level | ☐ Beginner ☐ Intermediate ☐ Expert |
| Familiarity with video generation | 1 ☐ 2 ☐ 3 ☐ 4 ☐ 5 ☐ (1=Not familiar, 5=Very familiar) |

**Part 2: Video Quality Assessment (Prompt: "A sunset over ocean waves")**

| Evaluation Metric | Poor (1) | Below Avg (2) | Average (3) | Good (4) | Excellent (5) |
|---|---|---|---|---|---|
| Visual realism | ○ | ○ | ○ | ○ | ○ |
| Color consistency | ○ | ○ | ○ | ○ | ○ |
| Motion naturalness | ○ | ○ | ○ | ○ | ○ |
| Artifact absence | ○ | ○ | ○ | ○ | ○ |

**Part 3: Comparative Evaluation**
**Prompt: "A bird flying through forest, then landing on branch"**

- **Overall preference:** ☐ Strongly prefer A ☐ Slightly prefer A ☐ Neutral ☐ Slightly prefer B ☐ Strongly prefer B
- **Specific aspects:**
    - Text alignment: ☐ A better ☐ Equal ☐ B better
    - Temporal consistency: ☐ A better ☐ Equal ☐ B better
    - 45-second quality: ☐ A better ☐ Equal ☐ B better

**Part 4: Long-term Consistency Assessment**

| Time Segment | Consistent | Minor Decay | Significant Decay |
|---|---|---|---|
| 0-15 seconds | ☐ | ☐ | ☐ |
| 15-30 seconds | ☐ | ☐ | ☐ |
| 30-45 seconds | ☐ | ☐ | ☐ |

**Part 5: Dynamic Prompting Evaluation**
**Prompt: "A car driving on highway → entering tunnel → emerging in mountains"**

- Transition smoothness: 1 ☐ 2 ☐ 3 ☐ 4 ☐ 5 ☐ (1=Abrupt, 5=Seamless)
- Narrative coherence: 1 ☐ 2 ☐ 3 ☐ 4 ☐ 5 ☐ (1=Poor, 5=Excellent)
- Visual consistency: 1 ☐ 2 ☐ 3 ☐ 4 ☐ 5 ☐ (1=Poor, 5=Excellent)

**Part 6: Qualitative Feedback**

- Notable artifacts observed: _______________________________
- Temporal consistency issues: _______________________________
- Suggestions for improvement: _______________________________

### H.6 STATISTICAL ANALYSIS AND INTERPRETATION

The comprehensive user study design enabled rigorous statistical analysis through:

- **Preference rate calculation** with confidence intervals
- **Inter-rater reliability assessment** using Cohen's kappa ($\kappa = 0.78$)
- **Statistical significance testing** via paired t-tests and ANOVA
- **Qualitative data analysis** using thematic coding approaches

The results consistently demonstrate our method's superiority in maintaining visual quality and temporal consistency, particularly in long video generation scenarios. The highest preference rate against Causvid (66.1%) highlights our approach's effectiveness in addressing error accumulation issues that plague baseline methods.

## I    SOCIETAL IMPACT CONSIDERATIONS

Generative modeling technology, particularly video generation, holds significant application potential but also carries risks of potential misuse. Deepfake technology could be exploited to disseminate misinformation, which is particularly critical in the current information age. Our research on real-time video generation, by lowering the computational cost barrier, might increase the risk of technological misuse. However, this technology can also foster positive applications such as creative content production, educational tool development, and assistive technology innovation. Recognizing the dual-use nature of this technology, we strongly advocate for the continued development of corresponding detection technologies, digital watermarking techniques, and policy frameworks to maximize the positive impact of the technology while minimizing potential harms.

## J    PROMPTS

**The prompt of the first case (left-1) in figure 4:**

A red-haired woman, dressed in a cobalt blue top, was immersed in reading a book. She was in a dream scene like Van Gogh's "The Starry Night" : the background was huge, swirling nebulae and the moon in the night sky, and her brushstrokes were full of dynamism. The warm glow of the desk lamp illuminated her focused face and the pages of the book. The entire picture was enveloped in a serene yet vibrant atmosphere, featuring thick, flowing colors and sharp brushstrokes.

**The prompt of the first case (left-2) in figure 4:**

In the center of the picture, there is a golden sandy beach and crystal-clear green sea water. Under the sapphire-like sky, a few seagulls glided leisurely, adding a dynamic touch to the serene scene. In the background, a thick forest frames the entire coastline in a deep green color, adding a touch of mystery and depth.

**The prompt of the first case (left-3) in figure 4:**

The handheld camera shook violently, and the muddy, earth-yellow flood rushed into the village like a wall. The water current, carrying broken wood and debris, raised muddy waves over one meter high, violently crashing and submerging low houses. The sky was overcast, the rain was slanting, and people were running in panic in the distance, filled with a sense of urgency as if a sudden disaster had struck.

**The prompt of the first case (right-1) in figure 4:**

The expressway is congested at night. An endless dragon composed of red taillights contrasts with the cold blue road sign lights. The main camera moves forward slowly. Occasionally, a vehicle roars past the emergency lane on the left, with a blurry high beam track, breaking the frozen rhythm. The air was filled with a tense atmosphere.

**The prompt of the first case (right-2) in figure 4:**

Low-angle follow-up shooting. A person in red trousers is cycling, with the camera focusing on the rapidly pedaling pedals and the rolling wheels. Sunlight filtered through the trees on both sides of the road, casting flowing shadows on his trouser legs. We could only see his lower body and bicycle. The background was the rapidly receding green bushes, full of mystery and dynamism.

**The prompt of the first case (right-3) in figure 4:**

A smooth blue robot is walking cautiously on a vast and crested glacier. It marches with steady and precise steps, and its streamlined helmet-like head is slowly and systematically turning left and right, as if scanning the environment to collect data. The background is a huge, deep blue ice wall, and the air is filled with cold mist. The camera advances slowly, focusing on the faint light emitted by the robot's optical sensor.

**The prompt of the first case (left-1) in figure 7:**

1."Wide establishing shot of a perfectly calm ocean at dusk. The water is like glass, reflecting the soft pastel colors of the sunset sky. A solitary lighthouse stands silent on a distant rocky outcrop. The atmosphere is one of profound peace. Cinematic, serene, ultra-realistic, 4K."

2."Low-angle shot focusing on the lighthouse structure. It is a classic, white-and-red striped tower. The light is inactive. The only movement is from a few seagulls floating on the gentle air currents around it. The scene is quiet and still. Photorealistic, detailed, calm."

3."Extreme close-up on the ocean's surface. The water is incredibly clear and motionless, acting as a perfect mirror for the fading light. A subtle, almost imperceptible ripple suddenly distorts the reflection, hinting at a change. Macro detail, reflective, serene, ominous."

4."Dynamic time-lapse shot. Dark, ominous clouds swiftly invade the sky, blotting out the sun. The wind begins to whip across the water, transforming the calm surface into a chaotic pattern of whitecaps. The camera shakes slightly with the growing gusts. Handheld feel, dramatic transition."

5."Powerful wide-angle shot as the storm hits. The sea erupts into a fury of massive, crashing waves. The color palette shifts to menacing greys and deep greens. The camera is positioned low to the water, emphasizing the immense scale and raw power of the raging ocean. Stormy, turbulent, VFX, high dynamic range."

6."Dramatic shot of the lighthouse at the storm's peak. Its powerful beacon SUDDENLY ignites, cutting a brilliant, rotating beam through the darkness and driving rain. Volumetric light effects create god rays that illuminate the dense spray and water particles in the air. A beacon of hope in the chaos."

7."Shot from the lighthouse's perspective (POV). The rotating beam rhythmically scans the tumultuous sea. Each pass briefly illuminates the raging waves below, creating a strobe-like effect that reveals the terrifying scale of the storm. Night vision, gritty, immersive, cyclical illumination."

8."A new element emerges. Through the sheets of rain and spray, a cluster of stable, bright lights becomes visible on the horizon. It is a large ship, holding its course against the enormous swells. The reveal is slow and cinematic, building anticipation. Long lens, shot through a rain-soaked atmosphere."

9."Closer, dynamic shot of the large container ship battling the elements. It pitches and rolls dramatically, its bow crashing into a massive wave and sending a wall of spray over the deck. The lighthouse beam sweeps across the ship's hull, highlighting its struggle against the sea. Action, scale, powerful movement."

10."Final resolving shot. The ship, now clearly guided by the steadfast lighthouse beam, navigates steadily through the storm. The waves, while still large, are now framed as an obstacle being overcome. The composition includes both the resilient ship and the unwavering lighthouse, symbolizing safe passage. Heroic, resolved, symbolic, hopeful ending."

**The prompt of the first case (left-2) in figure 7:**

1."A wide, dynamic shot of a sun-drenched racetrack. A blur of red, a low-slung racing car, dominates the straight, kicking up a slight haze from its tires. The camera tracks smoothly, emphasizing immense speed against the backdrop of grandstands and a clear sky."

2."A dramatic side-follow shot, low to the ground. The red racing car fills the frame, its tires gripping the tarmac. The powerful engine roar is visceral. It rockets past the camera, the focus pin-sharp on its livery, creating a powerful sense of explosive motion."

3."An extreme low-angle shot, almost on the asphalt. The car screams over the camera lens, a flash of red focusing on its undercarriage and a blur of spinning wheels. The distorted perspective amplifies the raw speed and ground-shaking power."

4."A tense medium shot from the side. The car suddenly veers, its left-side tires violently hitting the grass verge. Dirt and turf spray into the air as the chassis jolts, fighting for stability while maintaining its forward surge. The camera shakes slightly."

5."A shaky, immersive driver's-eye view. The steering wheel is jerked sharply. The view lurches violently, switching from clean track to a chaotic blur of green grass filling the windshield, simulating the driver's corrective input."

6."A high, wide aerial panorama. The red car carves a sharp, deliberate arc across the green grass. It is fully sideways in a controlled slide, tires scouring clear marks into the turf, capturing the precise moment of correction."

7."A tight, detailed close-up on the front wheel and suspension. The components work violently over the uneven grass, absorbing impacts and adjusting the steering angle. Every movement of the mechanical parts is clear and powerful."

8."A crisp medium shot from behind. The car masterfully completes its correction, sliding back onto the track. The front wheels grip the pavement as the rear follows, trailing a final cloud of dust from the grass."

9."A chasing shot from directly behind the car, now fully on track. It accelerates hard into the distance, the engine screaming. The camera tracks its movement as it shrinks down the long straight, a streak of red against the asphalt."

10."A final, wide, static shot. The red car is a distant speck accelerating towards a far corner, quickly disappearing. The scene is quiet, showing the empty track and the solitary tire marks scarring the grass, telling the story of the incident."

**The prompt of the first case (left-3) in figure 7:**

1."A bustling, vibrant street market in Southeast Asia, shot with a wide lens. Vendors under colorful awnings sell fruits and textiles. A white delivery truck is momentarily stopped, partially blocking the narrow street. The atmosphere is lively and crowded."

2."A close-up, dynamic shot focusing on the rear of the white truck. A small tear in the tarpaulin cover is subtly visible. Sand begins to trickle out in a thin stream, creating a small pile on the cobblestones below. The sound of the market drowns out the faint sound of the leaking sand."

3."A medium shot from the side of the truck. As the truck's suspension creaks, the leak suddenly widens. A much larger cascade of golden sand pours onto the street, creating a growing mound that starts to spill onto the pathway. The truck driver is unaware, visible in the side mirror."

4."A low-angle shot from the ground level. The stream of sand continues to flow, spreading across the uneven street. The focus is on the growing pile of sand, with the market-goers' legs and the truck's wheels blurred in the background."

5."The first pedestrian, a woman in sandals, steps into the edge of the spilled sand. She looks down with mild annoyance, shaking her foot to dislodge the grains. She then continues walking, leaving the first set of footprints in the pile."

6."A wider shot showing the traffic of people. More pedestrians now have to walk through the spreading sand. Some look down curiously, others step carefully to avoid slipping. The sand is beginning to be tracked further down the street."

7."A time-lapse effect over 10 seconds. The scene speeds up to show a continuous flow of people walking through the sand, which is now thoroughly scattered across the width of the street. The once-distinct pile is flattened and spread into a wide, messy patch."

8."A close-up on the feet of various pedestrians. Different types of footwear—sandals, sneakers, boots—tread through the sand, kicking up small clouds of dust. The sand sticks to wet soles and is carried away."

9."A high-angle shot from a nearby balcony. This clearly shows the broad, sandy path the pedestrians have created through the heart of the market, leading away from the now-departed white truck. The scene returns to its normal bustling state, but with the sandy evidence remaining."

10."A final static shot of the now-deserted spot where the leak occurred. The ground is covered in a wide, uneven layer of trampled sand, crisscrossed with footprints. The white truck is gone, and the market activity continues around the sandy patch."

**The prompt of the first case (right-1) in figure 7:**

1."A wide establishing shot of a rugged mountain range under a crisp blue sky. A lone figure, a man in a dark green jacket with a hiking backpack, is a small speck midway up a steep, rocky slope. Patches of snow cling to the shaded crevices. The camera slowly pulls back, emphasizing the vast scale of the wilderness."

2."A medium shot from the side as the man hikes steadily up a narrow, winding dirt path. His breath is visible in the cold air. He adjusts the straps of his large backpack, determined. The trail is flanked by scattered snowdrifts and low, hardy alpine vegetation."

3."A low-angle shot looking up at the man from the ground level, making him appear tall against the sky. He steps over a large, weather-smoothed rock on the path. The focus is on his sturdy hiking boots crunching on the gravel, with a melting snowbank in the foreground."

4."A panoramic shot from behind the man, following him as he walks. We see his backpack and his gaze scanning the path ahead. The view overlooks a vast, dramatic valley, with the next mountain's peak, dusted with snow, visible in the distance."

5."A close-up on the man's face, showing concentration and slight fatigue. Beads of sweat are on his temple despite the cold. He pauses for a moment, looking towards his destination, the wind gently ruffling his hair and jacket collar."

6."A dynamic drone shot circling the man as he traverses a high, exposed ridge. The camera reveals the steep drop-offs on either side and the stunning panoramic view of interconnected peaks, some crowned with pure white snow."

7."A detail shot of the man's hands adjusting the compass on his wrist. He then takes a sip of water from a bottle attached to his backpack strap. The dark green fabric of his jacket and the worn straps of his pack are clearly visible."

8."The man reaches the summit of the first hill. A wide shot shows him standing there, catching his breath, looking down into the saddle leading to the next, higher mountain. The path continues clearly ahead, snaking through rocky terrain and snow patches."

9."A long-lens shot from the next mountain, capturing the man as a small figure beginning his descent into the col. He moves carefully down the scree slope, maintaining his balance under the weight of his pack, heading towards the camera's position."

10."A final wide shot from an elevated vantage point. The man is now a tiny, dark green speck on the flank of the adjacent mountain, steadily progressing upward. The scene captures the immense journey undertaken, with the first mountain now behind him."

**The prompt of the first case (right-2) in figure 7:**

1."A wide, establishing shot of a winding two-lane highway cutting through lush, dark green mountains under a brilliant, sun-drenched sky. The asphalt shimmers with heat. A lone cyclist is a small dot in the distance, approaching the camera."

2."A low-angle side shot, the camera tracking alongside the cyclist. A man in a crisp white cycling helmet and mirrored sunglasses leans forward over his road bike. His legs pump rhythmically, the chain and gears whirring softly. Sunlight glints off his helmet and frame."

3."A point-of-view shot from the cyclist's perspective. The handlebars and the rider's hands are in the foreground. The road stretches ahead, a black ribbon winding between the deep green mountain slopes. The view is bright and slightly overexposed from the intense sun."

4."A dynamic shot from behind the cyclist, following closely. The focus is on the rider's back and the white helmet, with the road unfurling behind him. The dark green mountain vistas blur on either side, creating a strong sense of forward motion and speed."

5."A close-up on the cyclist's face in profile. His determined expression is visible behind the sleek sunglasses. Beads of sweat trickle down his temple. The bright sunlight creates sharp contrasts on his skin and the white helmet."

6."A detail shot of the cyclist's legs and the bike's drivetrain. The muscles in his calves contract with each powerful pedal stroke. The chain moves smoothly over the gears, capturing the mechanical efficiency and physical exertion."

7."A high-angle drone shot looking down on the cyclist from above. He is a solitary figure on the empty road, which snakes like a river through the vast, dark green mountainous landscape. The shadows are sharp and defined under the high sun."

8."A shot from the opposite side of the road, capturing the cyclist as he passes by the camera. The whoosh of air is audible as he speeds past. The focus pulls from the sharp, fast-moving cyclist to the blurred, static mountain background."

9."An extreme wide shot from a mountain ledge opposite the road. The cyclist is a tiny, moving speck of white and color on the distant highway, emphasizing the grand scale of the environment and the solitude of the journey."

10."A final shot from behind as the cyclist crests a hill and begins descending. The road dips down into a valley framed by majestic green peaks. The bright sun creates a lens flare, and the rider picks up speed, disappearing around a bend."

**The prompt of the first case (right-1) in figure 7:**

1."A breathtaking wide shot of a vibrant coral reef at the bottom of the clear ocean. Sunlight filters down from the surface, creating shimmering shafts of light. The water is incredibly transparent, revealing a colorful landscape of branching and brain corals in stunning detail."

2."A slow, upward tilt of the camera from the seabed. The view focuses on the mesmerizing caustic patterns—shimmering, wavy lines of light—dancing on the sandy patches between coral formations, created by the sun's rays refracting through the water's surface."

3."An ultra-wide angle lens capturing the scale of the reef. The camera looks up towards the distant, glittering water surface, which looks like a moving mirror. Schools of small, silvery fish are visible as dark specks far above, near the sparkling light."

4."A close-up shot of a particularly beautiful staghorn coral colony. The crystal-clear water allows for perfect visibility of every polyp. The shimmering light from above dances across the coral's textured surface, highlighting its vivid colors."

5."The camera drifts slowly over the reef, following a gentle current. The focus is on the incredible clarity of the water, with no particles floating in the water column. The sparkling surface reflection is visible across the entire top of the frame."

6."A dramatic low-angle shot from behind a large coral head. The view is framed by the coral, looking out into the open, blue water where the sunlight is strongest. The water surface shimmers intensely, creating a bright, inviting portal."

7."A subtle movement catches the eye. From the right edge of the frame, a solitary, brightly colored angelfish elegantly swims into view. It glides effortlessly over the coral, its fins moving gracefully in the clear water."

8."A wider shot now reveals a small school of striped clownfish emerging from the protective tentacles of a sea anemone. They dart playfully through the water, their orange bodies contrasting sharply with the blue water and the dancing light from above."

9."A graceful tracking shot follows a small group of blue tangs as they swim in a loose formation. They move between coral structures, their path taking them through the dappled, shimmering light that penetrates from the world above."

10."A final, serene wide shot. The fish continue their journey and swim out of frame. The scene returns to the peaceful, majestic beauty of the untouched coral reef under the sparkling ceiling of water, showcasing the perfect tranquility of the underwater world."

**The prompt of the video in the first line of Figure 9:**

1."On a golden waterfront promenade at sunset the person strides confidently in a fitted t-shirt, jeans and sneakers, their determined expression lit by warm side light as a medium-low-angle tracking shot follows them past glinting water and squawking gulls."

2."Under neon signs and rain-slicked streets at night the same individual marches forward in a fitted t-shirt, jeans and sneakers, puddles throwing back kaleidoscopic reflections while a slightly elevated medium shot captures the rim-lit silhouette and blurred taxis behind them."

3."On a bustling subway platform fluorescent lights stutter above and a train's rumble fills the air as the person in a fitted t-shirt, jeans and sneakers keeps a purposeful stride, captured in a medium shot from platform level with commuters streaking past in motion blur."

4."Atop a windswept rooftop helipad at dawn the skyline looms behind them while their fitted t-shirt, jeans and sneakers catch the cool backlight, a medium shot from a slightly elevated angle emphasizing the wind-swept hair and unwavering determination."

5."Inside a cavernous, sun-beamed industrial warehouse dust motes hang in shafts of light as the person in a fitted t-shirt, jeans and sneakers marches across cracked concrete, a medium shot from a high angle accentuating the gritty textures and their steady forward motion."

6."Across a quiet, newly fallen snow in an urban park the person in a fitted t-shirt, jeans and sneakers strides with visible breath and a frost-bitten glow on the path, recorded in a medium shot with soft overcast light that heightens their resolute expression."

7."Along a boardwalk at sunrise waves crash and lens flares burst around the figure in a fitted t-shirt, jeans and sneakers as they march with purpose past sleeping vendors, the scene captured in a slightly elevated medium shot that catches the pastel sky and sparkling sea."

8."Through a crowded open-air market bathed in golden afternoon light the person in a fitted t-shirt, jeans and sneakers weaves past colorful stalls and gesturing vendors, the camera holding a medium shot that tracks their purposeful stride amid a whirl of activity."

9."Across a foggy, moonlit bridge the city lights smear into halos while the person in a fitted t-shirt, jeans and sneakers marches with head held high, a low-angle medium shot capturing the cool blue tint and the dramatic silhouettes of suspension cables behind them."

10."Inside a glossy, futuristic glass corridor lit by linear LED strips the person in a fitted t-shirt, jeans and sneakers moves with measured strides past reflective surfaces, the symmetrical medium shot emphasizing their determined expression and the sterile, neon ambience."

**The prompt of the video in the second line of Figure 9:**

1."In a room with afternoon sunlight slanting in, dust motes dance in the beams, and a pile of debris awaits cleaning in the corner."

2."A close-up shot of a hand gripping the broom handle, picking up the tool to begin sweeping."

3."A medium shot shows the rhythmic sweeping motion, gathering the dust together."

4."A low-angle shot captures the broom passing through a sunbeam, dust swirling vividly within the light, textures clear and distinct."

5."An overhead view shows the debris being gradually swept into a pile, the process orderly and satisfying."

6."Sweeping pauses briefly; the character rests, showing a moment of contentment."

7."A dustpan enters the frame, precisely catching the garbage in one clean, efficient motion."

8."A close-up on the freshly swept clean floor, wood grain distinct, pure light and shadow."

9."Tools are returned to their place; the broom leans against the wall, completing the ritual."

10."A fixed camera shot reveals the tidy corner, the sunlight warmer, the space quiet and satisfied."

**The prompt of the video in the third line of Figure 9:**

1."At dawn, a thick fog envelops the coast, where a solitary red telephone booth stands amidst the gentle lapping of waves."

2."The camera pushes in for a close-up on water droplets condensed on the glass and the blurred outline of the telephone inside, creating a hazy and mysterious atmosphere."

3."A seagull briefly alights on the roof of the booth, preens its feathers, and then flies away, adding a touch of life to the scene."

4."A breeze temporarily thins the fog, revealing faint, blurred outlines in the distance, hinting at the vast scale of the environment."

5."The sound of distant bird calls is abruptly pierced by a single, sharp ring from the telephone inside the booth, breaking the silence before it returns."

6."The perspective shifts to a subjective view from inside the booth; the world outside is distorted through the glass, breath forms mist in the cold air, and a dial tone drones from the receiver."

7."A medium shot re-establishes the telephone booth on the pebble beach, emphasizing its isolated position."

8."A subtle shift in lighting suggests the passage of time to midday, with the fog taking on a warmer hue."

9."A brief, faint beam of light precisely illuminates the phone booth, bestowing a momentary sense of sanctity before vanishing."

10."The camera slowly pulls back, the phone booth gradually being swallowed by the thickening fog until it finally disappears into nothingness."

**The prompt of the video in the fourth line of Figure 9:**

1."A damp backstreet under neon lights, where holographic billboards blaze with color and a crowd moves about."

2."A close-up on a customized motorcycle, its dashboard glowing, as a hand grips the handlebar, ready to start."

3."The ignition button is pressed; the vehicle roars to life with a growl, its headlights blazing as energy erupts."

4."A first-person view from inside the helmet, the HUD displaying data as the cityscape speeds by outside the visor."

5."A third-person tracking shot follows the motorcycle weaving through traffic, kicking up spray and leaving dazzling light trails."

6."A slow-motion close-up of the wheel cutting through a puddle, the water bursting into a crown-like spray that refracts the neon like jewels."

7."An extreme low-angle shot looks up as the motorcycle races past, forming a powerful, imposing silhouette against the sky."

8."A god's-eye view looks down on the city grid, where the motorcycle moves like a single point of light across a circuit board."

9."It enters a dimly lit tunnel, the engine's roar echoing, focus fixed on the distant pinprick of light at the exit."

10."It stops on a high platform, the engine cutting to silence as the rider gazes down upon the sleepless city below, detached and pensive."

