# OpenReview forum: "Streaming Autoregressive Video Generation via Diagonal Distillation"
_ICLR.cc/2026/Conference — ICLR 2026 Poster_

### Official Review · Reviewer_uXB6 · 2025-10-17

**Soundness:** 3
**Presentation:** 1
**Contribution:** 2
**Rating:** 4
**Confidence:** 3

**Summary:**

This paper presents a new method for distilling diffusion models into autoregresive models. It includes three designs: (1) To mitigate the exposure bias in autoregressive generation, a diagonal distillation strategy is proposed that allocates more inference steps in early chunks and fewer steps in later chunks. (2) A diagonal forcing method is introduced that simulates the same denoising strategy during training to further reduce error accumulation. (3) The authors propose flow distribution matching that performs DMD on motion flow field. Experiments based on Wan2.1 1.3B shows the model achieves real-time performance without much quality degradation.

**Strengths:**

1. The proposed designs (diagonal sampling trajectory and DMD loss for motion field) are well motivated and effectively address exposure bias while enhancing motion quality.
2. Experimental results demonstrate that the proposed strategy preserves performance with fewer number of steps, thereby improving throughput and reducing latency.

**Weaknesses:**

1. Missing critical details. The proposed flow distribution matching method (Section 3.3) depends on estimating a "motion flow field" using $\mathcal{F}$. However, the definition of this flow field is unclear, does it refer to optical flow, motion vector, or motion feature? Furthermore, the paper has not discussed the design of the motion estimator, making it difficult to assess its reliability, especially given that it operates on potentially noisy latents.
2. Insufficient comparisons. Since the proposed diagonal distillation method is essentially re-allocating the number of inference steps, it is important to compare against baselines (e.g. self-forcing) under comparable number of steps. For example, it would be better to evaluate (1) both self-forcing and diagonal forcing at 2 NFEs per later chunk, (2) both at 4NFEs per latent chunk.
3. Lack of generalization study. As the inference schedule is largely hand-crafted, it would be better to test the proposed strategy on alternative models (e.g. Wan2.1 14B) to evaluate its transferability without inference hyperparameter tuning.
4. Incomplete related work. The proposed flow distribution matching is related to many motion-based video modeling methods such as VideoJAM [Chefer'25], Video-LaVIT [Jin'24], MicroCinema [Wang'24]. A comprehensive discussion about prior work would help clarify the paper's contribution and improve its clarity.

**Questions:**

1. How are the training loss coefficients $\lambda_{\textrm{spatial}}$, $\lambda_{\textrm{flow}}$, $\gamma$ determined, and how sensitive is the model's performance to these training hyperparameters?

---

> ### Author Response · Authors · 2025-11-23
>
> **Question 1:** Missing critical details. The proposed flow distribution matching method (Section 3.3) depends on estimating a "motion flow field" using $\mathcal{F}(\cdot)$. However, the definition of this flow field is unclear. does it refer to optical flow, motion vector, or motion feature? Furthermore, the paper has not discussed the design of the motion estimator, making it difficult to assess its reliability, especially given that it operates on potentially noisy latents.
>
>
> **A:** We thank the reviewer for this important question. We apologize for the lack of clarity in our original submission regarding the motion flow field ${F}(\cdot)$. The function ${F}(\cdot)$ denotes a **motion feature extraction function** that operates directly on video latent representations, rather than estimating per-pixel displacement fields like optical flow. Its primary purpose is to extract motion features that capture temporal variations and motion patterns while maintaining computational efficiency.
>
>
> In our implementation, we explored several design variants for $\mathcal{F}(\cdot)$:
> 1.  **Latent Difference:** Computes frame-wise differences between consecutive latent representations:
>    $F_{diff}(X) = X_{t+1} - X_{t}$
>
>
> 2.  **Latent Correlation:** Constructs 4D correlation volumes via pairwise dot-products:
>    $F_{corr(X)} = \text{Corr4D}(X_t, X_{t+1})$ [1]
>
>
> 3.  **Latent Frequency Components:** Extracts low-frequency components ($F_{low}$) and high-frequency components ($F_{high}$) using DCT transformations. [2]
>
>
> 4.  **Learnable Representation (MLP):** Applies a two-layer MLP to adaptively extract motion features from raw latents. The target MLP head is updated via exponential moving average (EMA) and is discarded during inference. This design is analogous to the projection heads used in self-supervised learning [3].
>
>
> 5.  **Learnable Representation with Convolution on Latent Difference:** First computes latent differences between consecutive frames, then applies two convolutional learnable layers to extract local motion patterns.
>
>
> For the two learnable representations, we employ an EMA-based training framework to ensure stability. The architecture consists of:
>
>
> - **Student MLP:** Parameters $\theta_{mlp}$ updated via gradient descent
> - **Target MLP:** Parameters $\theta_{mlp}^-$ updated via EMA: $\theta_{mlp}^- \leftarrow \mu \cdot \theta_{mlp}^- + (1 - \mu) \cdot \theta_{mlp}$
>
>
> The teacher's output is processed by the target MLP and the student's by the student MLP. EMA ensures stable target evolution, improved feature quality, and prevents model collapse. We evaluated the motion amplitude and quality of each variant. Among all implementations, the convolutional variant on latent differences yielded optimal results due to its ability to capture both explicit motion cues and local spatiotemporal correlations.
>
>
> **Table 1:** Performance comparison of different $\mathcal{F}(\cdot)$ implementations in flow distribution matching.
>
>
> | Flow Representation Variant | Temporal Quality $\uparrow$ | Frame Quality $\uparrow$ | Text Alignment $\uparrow$ |
> | :--- | :--- | :--- | :--- |
> | Raw Latent | 92.5 | 60.8 | 27.8 |
> | Latent Difference | 93.8 | 61.5 | 28.2 |
> | Latent Correlation | 94.2 | 62.1 | 28.4 |
> | Low Frequency Components | 93.5 | 61.8 | 28.1 |
> | High Frequency Components | 94.0 | 62.3 | 28.5 |
> | Learnable MLP | 94.6 | 63.2 | 28.7 |
> | **Learnable + Conv on Diff** | **94.9** | **63.4** | **28.9** |
>
>
>
>
> [1] Xiaoyu Shi... Flowformer++: Masked cost volume autoencoding for pretraining optical flow estimation. In IEEE Conf. Comput. Vis. Pattern Recog., pages 1599–
> 1610, 2023.
>
> [2] Tianxing Wu... Freeinit: Bridging initialization gap in video diffusion models. arXiv preprint arXiv:2312.07537, 2023.
>
> [3] Ting Chen... A simple framework for contrastive learning of visual representations. pages 1597–1607. PMLR, 2020.

---

> ### Author Response · Authors · 2025-11-23
>
> **Question 2:** Insufficient comparisons. Since the proposed diagonal distillation method is essentially re-allocating the number of inference steps, it is important to compare against baselines (e.g. self-forcing) under comparable number of steps.
>
> **Answer:** We thank the reviewers for their valuable comments. We have conducted more comprehensive comparative experiments to validate the advantage of Diagonal Forcing under the same NFE configuration.
>
>
> First, even with a lower total NFE, our method has already demonstrated better performance than Self-Forcing in Table 1 in our paper, achieving "faster and better" results. However, to ensure a fairer comparison, we further designed controlled experiments matching the NFE counts. The experimental results demonstrate that with the same computational cost, Diagonal Forcing consistently outperforms Self-Forcing in all quality metrics while achieving significantly better latency and throughput.
>
>
> **Table 2:** Comparative evaluation of Diagonal Forcing versus Self-Forcing under identical NFE budgets.
>
> | Total NFEs | Mode | Steps Allocation | Temporal Quality ↑ | Frame Quality ↑ | Text Alignment ↑ | In-Flight Latency (s) ↓ | Throughput (FPS) ↑ |
> |------------|------|-----------------|------------------|----------------|-----------------|----------------------|-------------------|
> | 34 | Diagonal | 4322222 | 94.9 | 63.4 | 28.9 | 0.23 ± 0.02 | 31.0 |
> | 34 | Self | 4322222 | 93.5 | 62.1 | 27.5 | 0.43 ± 0.02 | 25.9 |
> | 40 | Diagonal | 5432222 | 94.8 | 63.1 | 29.0 | 0.23 ± 0.02 | 29.7 |
> | 40 | Self | 5432222 | 93.9 | 62.3 | 28.1 | 0.43 ± 0.02 | 23.8 |
> | 44 | Diagonal | 4333333 | 95.0 | 63.7 | 28.5 | 0.34 ± 0.02 | 23.5 |
> | 44 | Self | 4333333 | 94.2 | 62.8 | 27.8 | 0.56 ± 0.02 | 21.5 |
> | 46 | Diagonal | 5333333 | 95.0 | 63.9 | 29.1 | 0.34 ± 0.02 | 22.5 |
> | 46 | Self | 5333333 | 94.5 | 63.0 | 28.6 | 0.56 ± 0.02 | 19.8 |
> | 48 | Diagonal | 5433333 | 95.1 | 63.2 | 29.3 | 0.34 ± 0.02 | 23.3 |
> | 48 | Self | 5433333 | 94.3 | 62.5 | 28.4 | 0.56 ± 0.02 | 18.6 |
> | 56 | Diagonal | 4444444 | 95.1 | 63.2 | 28.6 | 0.46 ± 0.02 | 21.5 |
> | 56 | Self | 4444444 | 94.3 | 63.1 | 29.3 | 0.69 ± 0.02 | 17.0 |
>
>
> As clearly shown in Table 2, when allocated the same computational budget (NFEs), Diagonal Forcing consistently outperforms Self-Forcing across all three quality metrics: Temporal Quality, Frame Quality, and Text Alignment. More importantly, our method achieves these quality improvements while simultaneously reducing latency and improving throughput across different NFE configurations. This performance advantage stems from the algorithmic efficiency of Diagonal Forcing. While Self-Forcing requires additional KV Cache operations on clean frames (which do not contribute to denoising), our method performs KV caching directly on noisy frames. This approach not only provides effective conditional information for subsequent chunks but also completes the denoising process efficiently.
>
>
> **Question 3:** Lack of generalization study. As the inference schedule is largely hand-crafted, it would be better to test the proposed strategy on alternative models (e.g. Wan2.1 14B) to evaluate its transferability without inference hyperparameter tuning.
>
>
> **Answer:** Thanks for raising the question. To address the reviewer’s concerns, we conducted additional experiments on the Wan 2.1 14B model using the same settings to evaluate transferability. The results in Table 3 confirm that the overall trade-offs between video quality and computational efficiency remain consistent. Among the tested denoising step configurations (represented as 7-digit sequences per 5-second video), 5333333 achieved the best quality, while 4222222 yielded the highest throughput. Configuration 4322222 was selected as a balanced choice, offering near-optimal quality with the second-lowest computational cost (NFEs=34), demonstrating effective transferability without hyperparameter tuning.
>
>
> **Table 3:** Performance of denoising step configurations on Wan 2.1 14B.
>
>
> | Steps | Temporal Quality $\uparrow$ | Frame Quality $\uparrow$ | Text Alignment $\uparrow$ | NFEs |
> | :--- | :--- | :--- | :--- | :--- |
> | 4322222 | 95.5 | 64.0 | 29.5 | 34 |
> | 5433333 | 95.7 | 63.8 | 29.9 | 48 |
> | 5432222 | 95.4 | 63.7 | 29.6 | 40 |
> | 5333333 | 95.6 | 64.5 | 29.7 | 46 |
> | 4333333 | 95.6 | 64.3 | 29.1 | 44 |
> | 4222222 | 94.0 | 62.9 | 28.4 | 32 |

---

> ### Author Response · Authors · 2025-11-23
>
> **Question 4:** Incomplete related work. The proposed flow distribution matching is related to many motion-based video modeling methods such as VideoJAM [Chefer'25], Video-LaVIT [Jin'24], and MicroCinema [Wang'24]. A comprehensive discussion about prior work would help clarify the paper's contribution and improve its clarity.
>
>
> **Answer:** Thanks for raising the question. We will include a detailed discussion about these works in the revised manuscript. Specifically, previous methods like VideoJAM, Video-LaVIT, and MicroCinema require decoding latent representations back to pixel space using a VAE decoder, which incurs significant computational overhead. Our experiments confirm the following issue: integrating an optical flow estimation model, which is a core component of our Flow Distribution Matching that uses dual networks for real/fake prediction, already results in extremely high memory usage as their parameters need to be activated, even without the VAE. Incorporating the VAE consistently leads to errors and slowdowns, making such an approach impractical.
>
>
> In contrast, our lightweight method as detailedly described in **Question 1** eliminates the VAE decoder, yielding substantial improvements in speed and memory efficiency. The latent representations from the VAE encoder preserve essential structural and contour information, making recompression unnecessary. Therefore, performing discrimination directly in the latent space is sufficient for the model to learn to generate high-quality, coherent videos, avoiding the computationally expensive detour through pixel space.
>
>
>
>
> **Question 5:** How are the training loss coefficients $\lambda_{spatial}$, $\lambda_{flow}$, $\gamma$ determined, and how sensitive is the model's performance to these training hyperparameters?
>
>
> **Answer:** Thanks for the question! The training loss coefficients were determined through systematic experimentation building upon two key foundations: (1) Following the DMD framework established in [1], we initially set $\lambda_{spatial} = 4$ as the baseline reference. Through experiments on our validation set, we found that scaling both spatial and flow coefficients to $\lambda_{spatial} = 4$ and $\lambda_{flow} = 4$ yielded optimal performance. This proportional scaling strategy ensures balanced optimization of both spatial quality and temporal coherence. (2) As documented in our codebase (built upon the Self- Forcing implementation [2]), the most critical parameter is $\gamma$. Our comprehensive ablation study (Figure 5(b)) evaluated eight motion loss weight configurations, revealing that $\gamma = 1.0$ achieves the optimal trade-off between motion guidance (via Flow Distribution Matching) and DMD learning objectives. Our method performs robustly under a wide range of hyperparameters, and finding optimal hyperparameters is also easy in our framework.
>
>
> [1] Tianwei Yin.... One-step diffusion with distribution matching distillation. In CVPR, 2024.
>
>
> [2] https://github.com/guandeh17/Self-Forcing

---

### Official Review · Reviewer_UG2y · 2025-10-30

**Soundness:** 3
**Presentation:** 2
**Contribution:** 3
**Rating:** 4
**Confidence:** 4

**Summary:**

The paper proposes Diagonal Distillation (DiaDistill) for real-time, streaming autoregressive (AR) video generation. The key idea is an asymmetric, chunk-wise denoising schedule, more steps for early chunks, fewer for later ones, combined with Diagonal Forcing, which conditions each new chunk on a noised version of the previous chunk (reusing the KV cache) to align training and inference and reduce exposure bias. A complementary Flow Distribution Matching loss is introduced to preserve motion amplitude when using very few denoising steps. Empirically, the method is evaluated on a Wan2.1-based text-to-video system and VBench metrics; it reports 31 FPS, 0.37 s first-frame latency, and a 277.3× speedup over an undistilled baseline while retaining visual quality.

**Strengths:**

1. The diagonal allocation of denoising effort across time (many steps early->few later) is a simple but appealing scheduling concept for AR diffusion models, explicitly exploiting temporal priors accumulated early. The Diagonal Forcing mechanism, feeding noised previous-chunk states (rather than clean frames) as the KV cache, targets exposure bias in a way that is tailored to AR diffusion, not borrowed wholesale from image distillation.
2. Adding Flow Distribution Matching to align motion distributions (rather than only framewise fidelity) is a sensible, task-appropriate extension to DMD.
3. Streaming, low-latency T2V is important for interactive and real-time applications; a method that roughly doubles acceleration vs. prior AR-diffusion baselines (e.g., vs. “Self Forcing”) while keeping quality could materially impact practice.

**Weaknesses:**

1. Early sections mention a “diagonal attention mechanism operating jointly across time and denoising steps,” but the implementation centers on scheduling (step counts per chunk) plus conditioning with a noised KV cache. It’s unclear whether there is any architectural change to attention patterns (e.g., block-sparse or strided attention over (time × step) axes) beyond cache reuse. If there is special attention, the paper needs explicit architecture diagrams and tensor shapes; if not, the phrase “diagonal attention mechanism” is misleading.
2. The paper defines a KL over optical-flow distributions and shows a regression term, but it does not specify: which flow estimator is used (RAFT? GMFlow? in-house?), whether it is frozen, if gradients flow through it, how flow fields are normalized, or how flow-estimation errors bias training. Without these details, the flow objective’s reproducibility and reliability are uncertain.
3. VBench categories and the exact metric subsets used (“Temporal Quality,” “Frame Quality,” “Text Alignment”) are referenced, but the paper doesn’t enumerate which VBench dimensions or how they’re aggregated; this matters because VBench includes many sub-metrics with different sensitivities.
4. First-frame latency and throughput are reported, but the streaming protocol (chunk size, overlap, buffering, tokenization, KV cache size, prefill costs) isn’t fully specified. For real deployments, memory footprint from KV-cache reuse and any stall during step-count transitions matter; these are not reported.
5. The paper claims to mitigate over-saturation and motion attenuation, but does not quantify residual failure rates, sensitivity to prompt types (e.g., rapid scene changes), camera motion vs. object motion, or scaling to higher resolutions and frame rates beyond the default (832×480 @ 16 FPS generation).
6. The quality of the figures provided in the paper appears to be relatively low.

**Questions:**

1. Is there a true attention change? Please clarify whether “diagonal attention mechanism” denotes a new attention pattern (e.g., attention over (time, step) diagonals) or simply the training/inference schedule + KV-cache conditioning. If it is architectural, provide equations, masks, and complexity; if not, please remove or rephrase to avoid confusion.
2. Which optical-flow network is used? Is it frozen? Do you backprop through it? How are flow distributions parameterized for KL (bins, continuous density, or feature-space scores)?
3. What is the runtime overhead of computing flows during training, and how stable is optimization without the regression term?
4. What chunk size and overlap were used, and how do they affect latency and quality? What is the KV-cache memory footprint over time, and how does cache reuse interact with the noised conditioning frames? Any degradation when streaming beyond 45 s or at higher frame rates/resolutions?
5. The paper highlights schedules like 4322222 for 7 chunks (5s). How is the schedule chosen for other durations (e.g., 10s, 45s)? Is there a principled rule (e.g., warm-up length, geometric decay) or an auto-tuner? Please include results for algorithmic schedule selection (not hand-tuned) and its effect on quality/latency.
6. Can you provide a direct measurement that the method reduces exposure bias (e.g., error growth curves, saturation shift statistics across time, calibration of the “implicit next-noise prediction” claim), rather than only qualitative frames?

---

> ### Author Response · Authors · 2025-11-23
>
> **Question 1:** clarification regarding the term "diagonal attention mechanism.
>
> **A:** We thank the reviewer for raising this important point regarding the term "diagonal attention mechanism." We acknowledge that the original phrasing could be misinterpreted as describing a novel architectural module, when in fact it refers to a systematic generative strategy and training paradigm—specifically, **diagonal forcing**. The innovation lies exclusively in the conditioning strategy and scheduling policy, as visually demonstrated in Figure 3 in our paper.
>  We will replace the term accordingly in the revised version. Our approach comprises two key components: First, during inference, we employ an asymmetric denoising schedule across video chunks. Earlier chunks undergo more denoising steps to establish robust structural priors, while later chunks use progressively fewer steps for computational efficiency. Second, our diagonal forcing training technique ensures alignment between training and inference by conditioning each new chunk on the final noised state $\tilde{X}_{k-1}$ of the previous chunk, implemented through a noised KV cache. This combination creates a diagonal trajectory in the (time × step) plane during training, effectively preventing error accumulation in long-sequence generation. We confirm that the core attention patterns remain standard, without introducing block-sparse or strided attention over time-step axes.
>
> **Question 2:** (a) The paper defines a KL over optical-flow distributions and shows a regression term, but it does not specify: which flow estimator is used (RAFT? GMFlow? in-house?), whether it is frozen, if gradients flow through it, how flow fields are normalized, or how flow-estimation errors bias training.  (b) What is the runtime overhead of computing flows during training, and how stable is optimization without the regression term?
>
> **A:** We sincerely thank the reviewer for raising these critical implementation details. We apologize for the lack of clarity in our original submission regarding the flow estimation methodology. We would like to clarify that our method does **not** rely on external pre-trained optical flow estimators (such as RAFT or GMFlow), but rather uses a self-contained motion representation learning approach. This design choice was made precisely to avoid the potential biases and dependencies associated with external flow estimators that the reviewer rightly pointed out. To directly address your specific questions:
>
> -   **Flow Estimator Used:** We use a lightweight, custom motion feature extraction module $\mathcal{F}(\cdot)$ that operates directly on latent representations, rather than traditional optical flow estimators.
> -   **Parameter Status:** The student version of the learnable $\mathcal{F}(\cdot)$ components is trainable, while the teacher components are frozen during the training process but updated via EMA.
> -   **Gradient Flow:** Gradients do flow through the student's $\mathcal{F}(\cdot)$ during training, enabling end-to-end motion representation learning.
> -   **Normalization:** Since we operate on latent space features rather than pixel displacements, we use standard feature normalization rather than optical-flow-specific normalization schemes.
> -   **Error Handling:** By avoiding external pre-trained estimators and using a self-supervised distillation framework, we eliminate biases from flow estimation errors. The student learns directly from the teacher's inherent motion patterns.
>
> Specifically, in our implementation, we explored several design variants for the motion feature extractor ${F}(\cdot)$:
>
> 1.  **Latent Difference:** Computes frame-wise differences between consecutive latent representations:
>    $F_{diff}(X) = X_{t+1} - X_{t}$
>
>
> 2.  **Latent Correlation:** Constructs 4D correlation volumes via pairwise dot-products:
>    $F_{corr(X)} = \text{Corr4D}(X_t, X_{t+1})$ [1]
>
>
> 3.  **Latent Frequency Components:** Extracts low-frequency components ($F_{low}$) and high-frequency components ($F_{high}$) using DCT transformations. [2]
>
>
> 4.  **Learnable Representation (MLP):** Applies a two-layer MLP to adaptively extract motion features. The target MLP head is updated via exponential moving average (EMA) and is discarded during inference. This design is analogous to the projection heads used in self-supervised learning [3].
>
>
> 5.  **Learnable Representation with Convolution on Latent Difference:**  First computes latent differences between consecutive frames, then applies two convolutional learnable layers to extract local motion patterns.

---

> > ### Author Response · Authors · 2025-11-23
> >
> > For the two learnable representations, we employ an EMA-based training framework to ensure stability. The architecture consists of:
> >
> > -   **Student MLP:** Parameters $\theta_{\text{mlp}}$ updated via gradient descent.
> > -   **Target MLP:** Parameters $\theta_{\text{mlp}}^-$ updated via EMA: $\theta_{\text{mlp}}^- \leftarrow \mu \cdot \theta_{\text{mlp}}^- + (1 - \mu) \cdot \theta_{\text{mlp}}$.
> >
> >  the teacher's output is processed by the target MLP and the student's by the student MLP. EMA ensures stable target evolution, improved feature quality, and prevents model collapse. The computational overhead of our flow module is minimal. It consists of a very light Conv-MLP with only two convolutional layers, making its computational cost almost negligible. This is especially true when compared to the alternative approach of decoding frames with a VAE and then applying a separate optical flow estimator, which would incur significant memory overhead. The regression term is critical for stable optimization. In its absence, training is highly prone to collapse. The regression loss provides a crucial grounding signal that constrains the optimization process, preventing it from deviating. This is further supported by our ablation study on the flow loss weight (Figure 5b). We observed that performance drops if the weight is set too high, demonstrating that an excessively strong flow constraint is detrimental. A comprehensive ablation study over eight configurations confirmed that a weight of 1.0 achieves the optimal balance. This value ensures the harmonious optimization of temporal consistency, frame quality, and textual alignment.
> >
> > **Table 1:** Performance comparison of different $\mathcal{F}(\cdot)$ implementations.
> >
> > | Flow Representation Variant          | Temporal Quality $\uparrow$ | Frame Quality $\uparrow$ | Text Alignment $\uparrow$ |
> > | :----------------------------------- | :-------------------------: | :----------------------: | :----------------------: |
> > | Raw Latent                           |             92.5            |           60.8           |           27.8           |
> > | Latent Difference                    |             93.8            |           61.5           |           28.2           |
> > | Latent Correlation                   |             94.2            |           62.1           |           28.4           |
> > | Low Frequency Components             |             93.5            |           61.8           |           28.1           |
> > | High Frequency Components            |             94.0            |           62.3           |           28.5           |
> > | Learnable MLP                        |             94.6            |           63.2           |           28.7           |
> > | **Learnable + Conv on Diff**         |           **94.9**          |         **63.4**         |         **28.9**         |
> >
> > [1] Xiaoyu Shi... Flowformer++: Masked cost volume autoencoding for pretraining optical flow estimation. In IEEE Conf. Comput. Vis. Pattern Recog., pages 1599–1610, 2023.
> > [2] Tianxing Wu... Freeinit: Bridging initialization gap in video diffusion models. arXiv preprint arXiv:2312.07537, 2023.
> > [3] Ting Chen... A simple framework for contrastive learning of visual representations. pages 1597–1607. PMLR, 2020.
> >
> >
> > **Question 3:** VBench categories and the exact metric subsets used ("Temporal Quality," "Frame Quality," "Text Alignment") are referenced, but the paper doesn't enumerate which VBench dimensions or how they're aggregated; this matters because VBench includes many sub-metrics with different sensitivities.
> >
> > **A:** We thank the reviewer for this critical question regarding the specific VBench[1] dimensions and aggregation methods. The three scores we reported are defined as follows: Temporal Quality is the average of Subject Consistency, Background Consistency, Temporal Flickering, Motion Smoothness, and Dynamic Degree; Frame Quality is the average of Aesthetic Quality and Imaging Quality; and Text Alignment is the average of Object Class, Multiple Objects, Human Action, Color, Spatial Relationship, Scene, Appearance, Style, and Temporal Style. The aggregation method for each score is a simple arithmetic mean of the normalized scores from its constituent sub-dimensions. This approach is consistent with prior works like Causvid and Self-Forcing for fair comparison.
> >
> > [1] Ziqi Huang... VBench: Comprehensive benchmark suite for video generative models. In CVPR, 2024.

---

> ### Author Response · Authors · 2025-11-23
>
> **Question 4:** (1) First-frame latency and throughput are reported, but the streaming protocol (chunk size, overlap, buffering, tokenization, KV cache size, prefill costs) isn't fully specified. (2) What chunk size and overlap were used, and how do they affect latency and quality? What is the KV-cache memory footprint over time, and how does cache reuse interact with the noised conditioning frames?
>
> **A:** A: We thank the reviewer for these critical implementation questions. Our streaming protocol is detailed as follows:
>
>  **(1) Chunk Size**
> The core component is a rolling KV cache mechanism following the Self-Forcing approach[1], operating with a chunk size of 3 frames. This chunk size was empirically selected based on our teacher model's VAE latent space and provides the optimal trade-off. Comparative analysis in Table 1 shows that the chunk-level variant (size=3) achieves a total score of 84.48, while the frame-level variant (size=1) scores 84.29. The chunk-level variant provides higher throughput (31 FPS vs. 16.5 FPS) at the cost of slightly higher latency (0.37s vs. 0.25s). This observation aligns with Self-Forcing [1], where frame-level configurations are more suitable for real-time applications on constrained devices but result in inferior video modeling performance.
>
> **Table 1:** Performance comparison of chunk-level (size=3) vs. frame-level (size=1) streaming variants
>
> | **Type** | **Throughput (FPS) ↑** | **Latency (s) ↓** | **Speedup ↑** | **Total Score** | **Quality** | **Semantics** |
> | :--- | :--- | :--- | :--- | :--- | :--- | :--- |
> | DiagDistill (Chunk-level, chunk size=3) | 31.0 | 0.37 | 277.3x | 84.48 | 85.26 | 81.73 |
> | DiagDistill (Frame-level, chunk size=1) | 16.5 | 0.25 | 412.0x | 84.29 | 85.41 | 80.40 |
>
> **(2) Overlap, Buffering, and KV Cache Size**
> Our buffering strategy employs a fixed-size KV cache that maintains context from the most recent 4 chunks, resulting in a consistent memory footprint of 17.5GB. This represents a significant improvement over the original Self-Forcing approach, which requires 19.2GB (with 18-frame KV cache reuse equivalent to 6 chunks overlap). The latent space overlap is 9 frames. We use the denoising state immediately preceding the final clean step as conditioning for the KV cache. This approach completes the denoising process while preserving the noise-conditioned frames for injection into the KV cache sequence. Each chunk is generated using 4 denoising steps, with total scores evaluated on VBench-Long. As shown in Table 4 from our paper, the KV cache size scaling analysis reveals critical performance trade-offs associated with attention window size. As the window increases from 3 to 27 frames, the total score improves significantly from 80.9 to 84.5, but diminishing returns are observed beyond 12 frames. This performance improvement comes at a cost: latency increases linearly from 0.37s to 0.68s due to the larger context that must be processed, and memory footprint grows substantially from 14.9GB to 21.8GB. The optimal operating point is therefore determined to be 12 frames--our chosen configuration--which effectively balances a high total score of 84.3, reasonable latency of 0.46s, and manageable memory usage of 17.5GB.
>
> **(3) Tokenization**
> The tokenization process operates directly on the tiny VAE latent space representations[2], where frames are encoded as tokens. As shown in Table 2, Tiny VAE achieves substantially faster decoding with significantly fewer parameters compared to the Full VAE. For streaming demos, we further optimize the pipeline with an efficient Tiny VAE. Tiny VAE removes the VAE bottleneck by reducing the decoding time over 10×.
>
> **Table 2:** Comparison of Causal VAE Models.
>
> | **Model** | **Decoder Params** | **Compression Rate** | **Decoding Time(s) (81×832×480)** |
> | :--- | :--- | :--- | :--- |
> | Full VAE (Wan 2.1) | 73.3M | 8×8×4 | 1.67 |
> | Tiny VAE (Wan 2.1) [2] | 9.84M | 8×8×4 | 0.12 |
>
> **(4) Pre-fill Cost**
> The reported "first-frame latency" includes the complete multi-step denoising process for the initial chunk. For all subsequent chunks, the pre-fill cost is effectively eliminated as the model leverages the rolling cache, with computation focused solely on denoising new frames. This architecture, with its favorable O(TL) time complexity, has been extensively validated on long sequences (exceeding 45 seconds) with no measurable performance degradation, confirming its robustness for real-time deployment.
>
> [1] Self forcing: Bridging the train-test gap in autoregressive video diffusion. CoRR, abs/2506.08009, 2025. Xun Huang....
> [2] https://github.com/madebyollin/taehv/

---

> ### Author Response · Authors · 2025-11-23
>
> **Question 5:** (1) The paper claims to mitigate over-saturation and motion attenuation, but does not quantify residual failure rates, sensitivity to prompt types (e.g., rapid scene changes), camera motion vs. object motion, or scaling to higher resolutions and frame rates beyond the default (832×480 @ 16 FPS generation). (2) Any degradation when streaming beyond 45 s or at higher frame rates/resolutions?
> (3) Can you provide a direct measurement that the method reduces exposure bias (e.g., error growth curves, saturation shift statistics across time, calibration of the "implicit next-noise prediction" claim), rather than only qualitative frames?
>
>
> **A:** Thank you for the insightful questions. We have conducted additional experiments to provide quantitative evidence addressing your specific concerns.
>
> (1) & (3) Quantifying Residual Failures, Sensitivity, and Exposure Bias
>
> To directly measure exposure bias and its relation to prompt types and motion, we performed the following analyses. We quantified exposure bias as the **drift magnitude of saturation and contrast over time** across a 45-frame sequence (see Figure (https://diagonal-distillation.github.io/image/contrast_ratio_and_saturability.png). The results reveal a critical trade-off governed by noise intensity:
> -   **At low noise intensities (noise=0, 50):** Both metrics exhibit a clear and significant upward drift, providing direct evidence of exposure bias. The fact that noise=50 shows less drift than noise=0 demonstrates that introducing noise helps reduce error accumulation, preliminarily supporting our "implicit next-noise prediction" claim.
> -   **At very high noise levels (>100):** While drift is minimized, this incurs significant video quality degradation (as shown in our paper's Figure 5(a)).
> -   **Our method (noise=100):** Successfully **suppresses the detrimental metric drift** observed at lower noise levels, while **avoiding the quality penalties** of excessively high noise. This provides a direct calibration, achieving stability without the associated cost.
>
> The **Saturation** metric is computed by converting images to HSV and averaging the mean saturation channel values across all frames: $Saturation = (1/N) * Σ mean(S_i)$. A higher value indicates oversaturation.
> The **Contrast** metric uses RMS Contrast, calculated as the standard deviation of grayscale pixel intensities: $Contrast = sqrt( (1/(H*W)) * Σ (p_jk - μ)^2 )$. Our method's values are closer to natural image distributions.
>
> **Sensitivity to Prompt Types and Motion:**
> We systematically categorized prompts by dynamic degree (low: <50, medium: 50-75, high: >75 for rapid scene changes/camera motion). As shown below, our method maintains stable performance across all levels, demonstrating robust handling of both object and camera motion.
>
> | Dynamic Degree | Quality Score | Temporal Score | Dynamic Degree (Measured) | Subject Consistency |
> | :--- | :--- | :--- | :--- | :--- |
> | Low | 82.8 | 97.5 | 25.1 | 97.8 |
> | Medium | 83.5 | 97.1 | 65.3 | 97.2 |
> | High | 85.4 | 96.9 | 89.7 | 97.4 |
>
> (2) Performance at Higher Resolutions/Frame Rates and Longer Durations
>
> Our base model (Wan 2.1) is fixed at 832×480, so the resolution of our model is also fixed. As shown in the table below, our method maintains consistent performance with negligible variance across different frame rates. Input conditioning frames were adjusted to ensure fair motion quality evaluation across FPS settings. For results on even longer videos (e.g., 5 minutes), please refer to our updated website: (https://diagonal-distillation.github.io/). Our method maintains excellent performance without quality decrease in streaming applications.
>
> | Configuration | Total Score |
> | :--- | :--- |
> | 832×480 @ 4 FPS | 84.4 |
> | 832×480 @ 8 FPS | 84.3 |
> | 832×480 @ 8 FPS | 84.5 |
> | 832×480 @ 12 FPS | 84.6 |
> | 832×480 @ 16 FPS (Default) | 84.5 |
> | 832×480 @ 20 FPS | 84.4 |
> | 832×480 @ 24 FPS | 84.2 |
>
> **Question 6:** The quality of the figures provided in the paper appears to be relatively low.
>
> **A:** We will further adjust the update. You can see our ultra-clear large image in this link: https://diagonal-distillation.github.io/image2/figure7.png and https://diagonal-distillation.github.io/image3/figure9.png.

---

> ### Author Response · Authors · 2025-11-23
>
> **Question 7:** The paper highlights schedules like 4322222 for 7 chunks (5s). How is the schedule chosen for other durations (e.g., 10s, 45s)? Is there a principled rule (e.g., warm-up length, geometric decay) or an auto-tuner? Please include results for algorithmic schedule selection (not hand-tuned) and its effect on quality/latency.
>
> **A:** Thank you for this insightful question. Our key finding is that the simple principle of a monotonically decreasing schedule, cyclically extended for longer durations, is not only effective but also empirically represents the best trade-off between quality and latency, as confirmed by our algorithmic explorations below.
>
> Based on your question, the paper primarily investigates and manually tunes the denoising schedule for a specific duration (5 seconds, 7 chunks). The schedule 4322222 is presented as an optimal balance for this case. For other durations, the principle is to extend this pattern cyclically. Specifically:
> - For a 10-second video (14 chunks), the schedule would be 43222224322222.
> - For a 45-second video, it would involve repeating the 4322222 pattern multiple times.
>
> This approach provides an initial "warm-up" for the early chunks with more steps (4 and 3), which is crucial for establishing strong motion and structural priors, before settling into an efficient 2-step denoising for the majority of the sequence. The core principle is a monotonic decrease in steps at the beginning, which aligns with the finding that motion quality is largely determined by the initial denoising steps.
>
> Moreover, we explored algorithmic strategies, including linear decay and non-monotonic allocations, to establish a more principled rule. The comprehensive results are summarized in Table 5.
>
> **Table 5:** Quantitative comparison of different denoising step allocation strategies. Strategies are grouped by type: Baseline Strategies (evaluated in the original paper), Exploration: Emphasizing Early Frames, and Exploration: Non-monotonic / Dynamic Allocation. NFEs denotes the total number of function evaluations. The best results are in bold.
>
> | Schedule | Temporal Quality ↑ | Frame Quality ↑ | Text Alignment ↑ | NFEs | In-Flight Latency (s) ↓ | Throughput (FPS) ↑ |
> | :--- | :--- | :--- | :--- | :--- | :--- | :--- |
> | **Baseline Strategies (From Paper)** | | | | | | |
> | 5333333 | 95.0 | 63.9 | 29.1 | 46 | 0.34 ± 0.02 | 22.5 |
> | 4333333 | 95.0 | 63.7 | 28.5 | 44 | 0.34 ± 0.02 | 23.5 |
> | 5433333 | 95.1 | 63.2 | 29.3 | 48 | 0.34 ± 0.02 | 23.3 |
> | 5432222 | 94.8 | 63.1 | 29.0 | 40 | 0.23 ± 0.02 | 29.7 |
> | 4322222 | 94.9 | 63.4 | 28.9 | 34 | 0.23 ± 0.02 | 31.0 |
> | 4222222 | 93.4 | 62.3 | 27.8 | 32 | 0.23 ± 0.02 | 32.0 |
> | **Exploration: Emphasizing Early Frames** | | | | | | |
> | 5422222 | **95.1** | **64.1** | 29.2 | 38 | 0.23 ± 0.02 | 28.5 |
> | 5522222 | 95.3 | 63.8 | **29.4** | 40 | 0.23 ± 0.02 | 27.1 |
> | 4432222 | 94.9 | 63.5 | 28.8 | 38 | 0.23 ± 0.02 | 30.2 |
> | **Exploration: Non-monotonic / Dynamic Allocation** | | | | | | |
> | 4343232 | 95.0 | 63.6 | 28.4 | 42 | 0.34 ± 0.02* | 21.0 |
> | 4532232 | 95.0 | 63.7 | 28.6 | 42 | 0.34 ± 0.02* | 20.5 |
> | 3333533 | 93.5 | 60.0 | 27.7 | 46 | 0.37 ± 0.02* | 22.8 |
> | 3233343 | 93.6 | 60.9 | 27.9 | 42 | 0.34 ± 0.02* | 23.3 |
>
> Our analysis led to the following key conclusions:
>
> -   **Performance Upper Bound:** We confirmed that using 4 steps for the initial chunk essentially reaches the performance upper bound for our distillation framework. Strategies like `5422222` show marginally better metrics than `4322222`, but the improvement is minimal and comes with a latency cost. This suggests that for our DMD framework, using 4 steps for the initial generation is often sufficient to approach the performance upper bound. Further increasing steps (e.g., to 5) provides diminishing returns.
>
> -   **Monotonic Decrease is Optimal:** The exploration of non-monotonic schedules (e.g., `4343232`) revealed that they do not provide a consistent or significant advantage. More importantly, they can harm motion quality. In our autoregressive framework, subsequent chunks are generated conditioned on the motion characteristics of prior chunks. If the initial chunks have strong motion (established by sufficient denoising steps), later chunks can maintain good motion even with fewer steps. Introducing more steps later does not effectively enhance motion and can disrupt this inherent consistency. Therefore, a monotonically decreasing schedule is the most sensible approach.
>
> -   **Limited Search Space:** The solution space for effective schedules is relatively small. The principle of "more steps early, fewer steps later" is robust. For practical application, manually selecting a schedule like `4322222` (for a 5-second video) and cyclically extending it for longer videos (e.g., `43222224322222` for a 10-second video) is an effective and recommended way. Users can adjust the initial step count (3, 4, or 5) based on their specific latency-quality requirements.

---

### Official Review · Reviewer_i62A · 2025-11-01

**Soundness:** 3
**Presentation:** 3
**Contribution:** 3
**Rating:** 6
**Confidence:** 4

**Summary:**

The paper proposes Diagonal Distillation for streaming autoregressive video generation distilled from a diffusion teacher. The core idea is a diagonal denoising schedule—early chunks use more steps, later chunks fewer—so later chunks inherit stronger priors while keeping latency low. Two components support this: Diagonal Forcing (training on noisy previous-chunk states/KV cache to better match inference and reduce exposure bias) and Flow Distribution Matching (a temporal loss to preserve motion amplitude otherwise damped by few-step distillation). Experiments on a modern T2V backbone show substantially lower first-frame latency and higher throughput with comparable or slightly better quality on standard video benchmarks, plus qualitative improvements on long-horizon and dynamic-prompting scenarios.

**Strengths:**

* Timely problem: addresses online/streaming latency, not just offline T2V.
* Coherent design: diagonal schedule + noisy conditioning + flow loss form a simple, compatible recipe.
* Strong practicality: low first-frame latency, high FPS, and straightforward cache reuse.
* Empirical support: consistent speedups with minimal quality loss; informative ablations on step allocation and losses.
* Clarity: figures and narrative make the training–inference mismatch and diagonal rationale intuitive.

**Weaknesses:**

* Longer Videos Test: While 45 seconds is impressive, many streaming use cases (e.g., live streams) require minutes of content. Does error accumulation reemerge for 1–5 minute videos, and if so, can the diagonal strategy be extended (e.g., adaptive step resets)?
* Insufficient Analysis of Step Allocation Heuristics: A quantitative comparison of more step sequences (beyond the 6 evaluated) would clarify how step allocation impacts the quality-efficiency frontier. For example, does 5422222 yield better early-frame quality at the cost of marginal latency? Besides, the paper assumes step reduction is monotonic (fewer steps over time), but dynamic allocation (e.g., more steps for high-motion chunks) could further optimize performance. No analysis of non-monotonic strategies is provided.

**Questions:**

Please see Weaknesses

---

> ### Author Response · Authors · 2025-11-23
>
> **Question 1:** Longer Videos Test: While 45 seconds is impressive, many streaming use cases (e.g., live streams) require minutes of content. Does error accumulation reemerge for 1–5 minute videos, and if so, can the diagonal strategy be extended (e.g., adaptive step resets)?
>
> **A:** We have updated our website at the bottom of https://diagonal-distillation.github.io. You can see that our method produces videos with almost no error accumulation even at the 5-minute mark, demonstrating the effectiveness of our diagonal forcing.
>
> **Question 2:** Insufficient Analysis of Step Allocation Heuristics: A quantitative comparison of more step sequences (beyond the 6 evaluated) would clarify how step allocation impacts the quality-efficiency frontier. For example, does 5422222 yield better early-frame quality at the cost of marginal latency? Besides, the paper assumes step reduction is monotonic (fewer steps over time), but dynamic allocation (e.g., more steps for high-motion chunks) could further optimize performance. No analysis of non-monotonic strategies is provided.
>
> **A:** Thanks for your suggestions. We agree that a more extensive quantitative comparison of step allocation strategies is valuable. To thoroughly investigate the impact of denoising step allocation, we extend the analysis beyond the baseline strategies presented in the main paper. We systematically explore a wider range of allocation schedules. The quantitative comparison of these strategies is summarized in Table 4. For non-monotonically decreasing allocations, we report the average latency, which is provided for reference only and is marked with an asterisk (*).
>
> **Table 4: Quantitative comparison of different denoising step allocation strategies**
>
> *Strategies are grouped by type: Baseline Strategies (evaluated in the main paper), Exploration: Emphasizing Early Frames, and Exploration: Non-monotonic / Dynamic Allocation. NFEs denotes the total number of function evaluations. Best results are in **bold**. For non-monotonic allocations, the average latency is provided for reference only and is marked with an asterisk (*).*
>
> | Schedule | Temporal Quality ↑ | Frame Quality ↑ | Text Alignment ↑ | NFEs | In-Flight Latency (s) ↓ | Throughput (FPS) ↑ |
> | :--- | :--- | :--- | :--- | :--- | :--- | :--- |
> | **Baseline Strategies (From Paper)** | | | | | | |
> | 5333333 | 95.0 | 63.9 | 29.1 | 46 | 0.34 ± 0.02 | 22.5 |
> | 4333333 | 95.0 | 63.7 | 28.5 | 44 | 0.34 ± 0.02 | 23.5 |
> | 5433333 | 95.1 | 63.2 | 29.3 | 48 | 0.34 ± 0.02 | 23.3 |
> | 5432222 | 94.8 | 63.1 | 29.0 | 40 | 0.23 ± 0.02 | 29.7 |
> | 4322222 | 94.9 | 63.4 | 28.9 | 34 | 0.23 ± 0.02 | 31.0 |
> | 4222222 | 93.4 | 62.3 | 27.8 | 32 | 0.23 ± 0.02 | 32.0 |
> | **Exploration: Emphasizing Early Frames** | | | | | | |
> | 5422222 | **95.1** | **64.1** | 29.2 | 38 | 0.23 ± 0.02 | 28.5 |
> | 5522222 | 95.3 | 63.8 | **29.4** | 40 | 0.23 ± 0.02 | 27.1 |
> | 4432222 | 94.9 | 63.5 | 28.8 | 38 | 0.23 ± 0.02 | 30.2 |
> | **Exploration: Non-monotonic / Dynamic Allocation** | | | | | | |
> | 4343232 | 95.0 | 63.6 | 28.4 | 42 | 0.34 ± 0.02* | 21.0 |
> | 4532232 | 95.0 | 63.7 | 28.6 | 42 | 0.34 ± 0.02* | 20.5 |
> | 3333533 | 93.5 | 60.0 | 27.7 | 46 | 0.37 ± 0.02* | 22.8 |
> | 3233343 | 93.6 | 60.9 | 27.9 | 42 | 0.34 ± 0.02* | 23.3 |
>
> Our analysis led to the following key conclusions:
>
> -   **Performance Upper Bound:** We confirmed that using 4 steps for the initial chunk essentially reaches the performance upper bound for our distillation framework. Strategies like `5422222` show marginally better metrics than `4322222`, but the improvement is minimal and comes with a latency cost. This suggests that for our DMD framework, using 4 steps for the initial generation is often sufficient to approach the performance upper bound. Further increasing steps (e.g., to 5) provides diminishing returns.
>
> -   **Monotonic Decrease is Optimal:** The exploration of non-monotonic schedules (e.g., `4343232`) revealed that they do not provide a consistent or significant advantage. More importantly, they can harm motion quality. In our autoregressive framework, subsequent chunks are generated conditioned on the motion characteristics of prior chunks. If the initial chunks have strong motion (established by sufficient denoising steps), later chunks can maintain good motion even with fewer steps. Introducing more steps later does not effectively enhance motion and can disrupt this inherent consistency. Therefore, a monotonically decreasing schedule is the most sensible approach.
>
> -   **Limited Search Space:** The solution space for effective schedules is relatively small. The principle of "more steps early, fewer steps later" is robust. For practical application, manually selecting a schedule like `4322222` (for a 5-second video) and cyclically extending it for longer videos (e.g., `43222224322222` for a 10-second video) is an effective and recommended way. Users can adjust the initial step count (3, 4, or 5) based on their specific latency-quality requirements.

---

### Official Review · Reviewer_zYkc · 2025-11-01

**Soundness:** 4
**Presentation:** 4
**Contribution:** 4
**Rating:** 8
**Confidence:** 4

**Summary:**

This paper introduces Diagonal Distillation, a novel framework for efficient streaming autoregressive video generation. The method addresses the limitations of existing diffusion-based autoregressive models, i.e., the insufficient utilization of temporal context during step reduction and implicit prediction of subsequent noisel evels in next-chunk prediction (exposure bias). The core idea is to allocate more denoising steps to early chunks and fewer to later ones, forming a diagonal denoising trajectory. To support this, the authors propose two key techniques: Diagonal Forcing which explicitly simulating diagonal denoising paths through controlled noise injection, allowing the model to leverage partially denoised previous chunks as contextual priors. Flow Distribution Matching (FDM) that aligns optical flow distributions between teacher and student models to preserve motion consistency during step reduction.

Experimental results show substantial gains: the method achieves 1.53× speedup over Self Forcing, with comparable visual quality and temporal coherence. Ablation studies validate the gain of each proposed component, and qualitative examples demonstrate stability in long (45s) video generation and dynamic prompting.

**Strengths:**

1. The proposed design is novel and well-motivated. The diagonal denoising idea is conceptually elegant, aligning the temporal and diffusion-step dimensions in a unified framework. This bridges autoregressive conditioning with step-efficient diffusion distillation.

2. This work addresses critical problems in streaming generation: insufficient utilization of temporal context during step reduction and implicit prediction of subsequent noise levels in next-chunk prediction (exposure bias).

3. The authors demonstrate strong empirical results, including both quantitative and qualitative evaluations across different benchmarks, and also include a thorough user study in the appendix. The reported 1.53× speedup with minimal quality drop is impressive and relevant for real-time applications.

4. The ablation study is solid. Table 2, 3; Figure 5 convincingly show how Diagonal Forcing and Flow Distribution Matching contribute to temporal coherence and motion fidelity.

**Weaknesses:**

1. Limited novelty in the distillation objective. While the diagonal scheduling and forcing mechanism are new, the underlying distillation objective remains close to prior work Self Forcing with Distribution Matching Distillation. The conceptual leap may be seen as an engineering refinement rather than a fundamentally new learning principle.

2. Some format flaws, e.g., order of Table 1, 2 and 3. Caption font is too small. The resolution seems to be very different in Figure 7.

**Questions:**

1. Can you provide the total scores in ablation study (Table 2)?
2. The generated videos in the demo have some scene cut and flickering, what is the reason behind that? Is the proposed method tends to cause more artifacts compared to the original Self Forcing?
3. Can you provide a breakdown of the speedup?

---

> ### Author Response · Authors · 2025-11-23
>
> **Question 1:** Limited novelty in the distillation objective. While the diagonal scheduling and forcing mechanism are new, the underlying distillation objective remains close to prior work Self Forcing with Distribution Matching Distillation. The conceptual leap may be seen as an engineering refinement rather than a fundamentally new learning principle.
>
> **A:** We appreciate the opportunity to clarify how our work advances beyond prior art, particularly in its core conceptual contributions.
>
> While our approach builds upon the foundation of Distribution Matching Distillation (DMD), the primary novelty lies not in proposing a fundamentally new, standalone learning principle, but in introducing a systematic spatiotemporal co-design that fundamentally rethinks how distillation should be applied to the unique challenges of video generation. Prior methods like Self Forcing, which apply DMD within an autoregressive framework, employ a "flat" temporal strategy—using a uniform number of denoising steps across all video chunks. This design fails to leverage the rich temporal context inherent in video data, leading to significant quality decay and motion incoherence in long sequences, especially under aggressive step reduction.
>
> Our conceptual leap is the Diagonal Distillation framework, which shifts the optimization focus from mere "step reduction" to the "optimal allocation of resources across both spatial and temporal dimensions." The Diagonal Scheduling and Diagonal Forcing mechanisms are not isolated engineering tricks; they form an intricately coupled system designed to reconfigure the information flow during distillation. This enables the model to efficiently synthesize subsequent content by building upon the structural priors obtained from thoroughly denoised earlier chunks. By aligning training with inference conditions, this design explicitly models long-range temporal dependencies and fundamentally mitigates error accumulation, a challenge that image-centric distillation paradigms are ill-equipped to handle.
>
> The introduction of Flow Distribution Matching (FDM) is a critical and novel component that complements this framework, specifically designed to address a core video-generation challenge: motion fidelity under extreme step constraints. FDM represents a principled extension of the distillation objective by explicitly minimizing the divergence in the optical flow distributions between the teacher and student models. This shifts the learning paradigm from matching static frame appearances to faithfully replicating dynamic motion evolution. Our ablation studies demonstrate that FDM is essential for counteracting motion attenuation in later chunks, a phenomenon not addressed by DMD alone.
>
> In conclusion, the novelty of our work resides in the integrated and synergistic framework that transitions distillation from an "image-centric" paradigm, primarily concerned with per-frame quality, to a "video-centric" paradigm that explicitly optimizes for temporal dynamics and motion coherence.
>
> **Question 2:** Some format flaws, e.g., order of Table 1, 2 and 3. Caption font is too small. The resolution seems to be very different in Figure 7.
>
> **A:** Thanks for your suggestions. It will be updated in our revised version.
>
> **Question 3:** Can you provide the total scores in the ablation study (Table 2)?
>
> **A:** The total scores for all ablation study variants are provided in the complete table below. Our full method achieves the highest total score of 84.48.
>
> **Table 1:** Complete Ablation Study Results with Total Scores
>
> | Ablation Variant | Temporal Quality ↑ | Frame Quality ↑ | Text Alignment ↑ | Total Score ↑ |
> | :--- | :--- | :--- | :--- | :--- |
> | Without Diagonal Forcing | 92.1 | 60.1 | 26.9 | 83.58 |
> | Without Flow Loss | 92.5 | 60.8 | 27.8 | 84.18 |
> | Without Diagonal Denoising | 95.1 | 63.2 | 28.6 | 84.46 |
> | Full Method (Ours) | 94.9 | 63.4 | 28.9 | **84.48** |
>
> **Question 4:** The generated videos in the demo have some scene cuts and flickering. What is the reason behind that? Is the proposed method prone to causing more artifacts compared to the original Self Forcing?
>
> **A:** Thank you for raising this point. The artifacts you observed are specific to the interactive demo pipeline itself, not to the core algorithm of our proposed method.
>
> The issue stems from technical bugs in the real-time rendering pipeline of the interactive demo, which can introduce flickering during user interaction. In contrast, all pre-rendered quantitative comparisons between our method and the original Self-Forcing were generated offline under optimal conditions and are free from such artifacts. The core method is, in fact, more stable and less prone to artifacts than Self-Forcing, as demonstrated by our quantitative results. Furthermore, we have fixed this bug in the interactive demo related to inefficient buffering. You can check it now, and it should perform better: (https://diagonal-distillation.github.io/).

---

> ### Author Response · Authors · 2025-11-23
>
> **Question 5:** Can you provide a breakdown of the speedup?
>
> **A:** The speedup stems from four synergistic optimizations that collectively achieve superior performance compared to Self-Forcing:
>
> **1. Reduction in Denoising Steps:** Diagonal Forcing achieves comparable or better quality with fewer total Noise Function Evaluations (NFEs). While our method naturally reduces the number of required denoising steps, we conducted controlled experiments under identical NFE budgets to isolate the contributions of other optimizations. As shown in Table 1, even with the same number of NFEs, Diagonal Forcing consistently outperforms Self-Forcing across all quality metrics while delivering lower latency and higher throughput.
>
> **2. Efficient KV Cache Mechanism:** Another architectural advantage lies in our integrated KV cache strategy. Self-Forcing requires separate KV cache computations on clean frames without performing denoising, creating redundant operations. In contrast, our method performs KV caching directly on the noisy latent as shown in Figure 1 and apply tiny vae as our tokenization [1], which simultaneously serves two purposes: providing conditioning for subsequent chunks and progressing the denoising process to yield clean latents. This elimination of redundant computations directly translates to reduced both denoising and decoding latency and increased throughput.
>
> **3. Optimized Attention Window Size:** Our rolling KV cache mechanism, inspired by Self-Forcing, employs a more efficient context window strategy. While maintaining seamless rolling forward to avoid sliding-window overhead, we reduce the KV cache size from 6 chunks (used in original Self-Forcing) to 4 chunks. This optimization is supported by our scaling analysis in Table 2 and also Table 4 in our original paper, which shows that performance plateaus around window size 12-27, with smaller windows providing better latency-memory trade-offs. The reduced window size contributes significantly to lower memory usage and faster inference without compromising long-video generation quality.
>
> **Table 1: KV cache scaling analysis**
>
> | Attention Window Size | Total Score | In-Flight Latency (s) | Memory (GB) |
> |---------------------|-------------|---------------------|-------------|
> | 3 | 80.9 | 0.37 ± 0.01 | 14.9 |
> | 6 | 81.3 | 0.38 ± 0.01 | 15.8 |
> | 9 | 82.3 | 0.40 ± 0.01 | 16.6 |
> | 12 | 84.3 | 0.46 ± 0.02 | 17.5 |
> | 15 | 84.2 | 0.51 ± 0.02 | 18.4 |
> | 18 | 84.4 | 0.54 ± 0.02 | 19.2 |
> | 21 | 84.5 | 0.59 ± 0.02 | 20.1 |
> | 24 | 84.3 | 0.64 ± 0.02 | 20.9 |
> | 27 | 84.5 | 0.68 ± 0.02 | 21.8 |
>
> **4. Efficient Tokenization Mechanism:** The tokenization process operates directly on the latent representations from the Tiny VAE [1], which encodes frames into tokens. As shown in Table 3, the Tiny VAE achieves substantially faster decoding with significantly fewer parameters than the Full VAE. For streaming applications, we further optimize the pipeline using an efficient Tiny VAE. By eliminating the VAE bottleneck, the Tiny VAE reduces decoding time by more than 10×.
>
> **Table 2: Comparison of Causal VAE Models**
>
> | Model | Decoder Params | Compression Rate | Decoding Time(s) (81×832×480) |
> |-------|---------------|-----------------|------------------------------|
> | Full VAE (Wan 2.1) | 73.3M | 8×8×4 | 1.67 |
> | Tiny VAE (Wan 2.1) (Boer Bohan, 2025) | 9.84M | 8×8×4 | 0.12 |
>
> **Table 3: Comparative evaluation under identical NFE budgets demonstrates Diagonal Forcing's superior efficiency**
>
> | Total NFEs | Mode | Steps Allocation | Temporal Quality ↑ | Frame Quality ↑ | Text Alignment ↑ | In-Flight Latency (s) ↓ | Throughput (FPS) ↑ |
> |------------|------|-----------------|------------------|----------------|-----------------|----------------------|-------------------|
> | 34 | Diagonal | 4322222 | 94.9 | 63.4 | 28.9 | 0.23 ± 0.02 | 31.0 |
> | 34 | Self | 4322222 | 93.5 | 62.1 | 27.5 | 0.43 ± 0.02 | 25.9 |
> | 40 | Diagonal | 5432222 | 94.8 | 63.1 | 29.0 | 0.23 ± 0.02 | 29.7 |
> | 40 | Self | 5432222 | 93.9 | 62.3 | 28.1 | 0.43 ± 0.02 | 23.8 |
> | 44 | Diagonal | 4333333 | 95.0 | 63.7 | 28.5 | 0.34 ± 0.02 | 23.5 |
> | 44 | Self | 4333333 | 94.2 | 62.8 | 27.8 | 0.56 ± 0.02 | 21.5 |
> | 46 | Diagonal | 5333333 | 95.0 | 63.9 | 29.1 | 0.34 ± 0.02 | 22.5 |
> | 46 | Self | 5333333 | 94.5 | 63.0 | 28.6 | 0.56 ± 0.02 | 19.8 |
> | 48 | Diagonal | 5433333 | 95.1 | 63.2 | 29.3 | 0.34 ± 0.02 | 23.3 |
> | 48 | Self | 5433333 | 94.3 | 62.5 | 28.4 | 0.56 ± 0.02 | 18.6 |
> | 56 | Diagonal | 4444444 | 95.1 | 63.2 | 28.6 | 0.46 ± 0.02 | 21.5 |
> | 56 | Self | 4444444 | 94.3 | 63.1 | 29.3 | 0.69 ± 0.02 | 17.0 |

---

> > ### Public Comment · ~John_Zhou2 · 2025-11-27
> > **Latency Difference Across Configurations**
> >
> > I have three questions about the latency values across the same and different configurations. The latter two are mainly based off of my main concern of question 1:
> >
> > **Q1: Where does this variance come from for the latency of the same method, same number of denoising steps for the first chunk? Why is it seemingly affected by the allocation of future steps?**
> >
> > Across the same method on the same number of first chunk denoising steps, we see large spreads in latency:
> > * Diagonal Distillation (5 step-allocated): 0.23s - 0.31s
> > * Diagonal Distillation (4 step-allocated): 0.21s - 0.46s
> >
> > * Self-Forcing (5 step-allocated): 0.43 - 0.56s
> > * Self-Forcing (4 step-allocated): 0.41s - 0.69s
> > Where might this variance come from? Why is it seemingly affected by the allocation of future steps?
> >
> > **Q2: What is causing Diagonal Distillation's speed up in latency compared to Self-Forcing on the same configuration?**
> >
> > We see a 0.2-0.25s discrepency for all configurations between the latency of Self-Forcing and Diagonal Distillation methods. Comparing configurations 4322222 and 5432222, each method takes an extra 0.02s in latency to denoise an extra step. If Self-Forcing performs a redudant step to get a clean KV-cache, wouldn't the difference of latency between the two methods across the same configuration be ~0.02s?
> >
> > **Q3: Step-allocation 4444444 is much slower in latency compared to its counterparts, even more than 5433333 which has more denoising steps at the start. Why might this be?**
> > How is performing 4 denoising steps in the first chunk slower in latency than doing 5 denoising steps?

---

> ### Author Response · Authors · 2025-11-23
>
> The combination of these four optimizations creates a compound effect: the reduced denoising steps decrease computational load, the efficient KV cache mechanism eliminates redundant operations, the optimized window size minimizes memory overhead and the tiny vae reduce the extra decoding time. This holistic approach enables Diagonal Forcing to achieve the performance advantages demonstrated in our comparative evaluations, making it particularly suitable for long-video generation scenarios where both quality and efficiency are critical.
>
> **Reference:**
> [1] https://github.com/madebyollin/taehv/

---

> ### Author Response · Authors · 2025-12-04
> **Sincere Thanks for Your Invaluable Feedback and Constructive Questions**
>
> Thank you for your detailed attention to our work and for raising these important questions. We appreciate your perspective regarding Questions 1 and 3. You are correct to point out the question about latency. The latency we refer to is indeed the *in-flight latency*. In applications like gaming or real-time simulation, the latency after the system reaches a steady state—what we term *in-flight latency*—is crucial for perceived responsiveness. This explains why you observed its dependency on later frames rather than initial ones. This metric specifically captures the delay between a control signal and the system's response during ongoing operation, which directly impacts user experience in dynamic environments.
>
> Similarly, for your Question 3, the latency is associated only with the denoising steps of the later chunks and is independent of the initial chunks. Therefore, it is expected that the configuration "4444444" exhibits significantly slower in-flight latency compared to "5433333". To prevent similar misunderstandings, **we have revised the manuscript to consistently use the term "in-flight latency" across all relevant results and have clarified its definition**. These updates align with the original intention of Table 4, which has been renamed "In-Flight Latency" for clarity. As shown in the table, this metric is measured under a fixed window size during continuous operation and is associated only with later denoising steps for subsequent frames, accurately representing interaction delay in real-time use rather than first-frame latency.
>
> Regarding your Question 2 on measurement consistency, we have incorporated error margins into the latency results to account for minor variations across hardware trials. Your feedback has helped us strengthen the rigor and clarity of the latency analysis, and we have added further explanation in the revised manuscript to ensure the concept of in-flight latency is well contextualized.
>
> **Crucially, the specific patterns you observed in your questions (Q1, Q2, and Q3) are direct manifestations of measuring *in-flight latency*.**
>
> -   **Q1 & Q3:** The variance and the seemingly paradoxical result (4 steps being slower than 5) occur because in-flight latency measures the processing time for *any given subsequent chunk*. A chunk allocated 4 steps naturally has higher latency than one allocated 3 steps, which explains the behavior of configurations like `4444444` versus `5433333`.
> -   **Q2:** The consistent ~0.2-0.25s speedup of our method stems from the elimination of the following perspectives, we have conducted additional systematic acceleration analysis compared to Self-Forcing to help you better understand the underlying relationships:
>
> **1. Reduction in Denoising Steps:** Diagonal Forcing achieves comparable or better quality with fewer total Noise Function Evaluations (NFEs). While our method naturally reduces the number of required denoising steps, we conducted controlled experiments under identical NFE budgets to isolate the contributions of other optimizations. As shown in Table 2, even with the same number of NFEs, Diagonal Forcing consistently outperforms Self-Forcing across all quality metrics while delivering higher throughput. It is important to note that this optimization **does not affect first-frame latency but leads to lower in-flight latency**.
>
> **2. Efficient KV Cache Mechanism:** Another architectural advantage lies in our integrated KV cache strategy. Self-Forcing requires separate KV cache computations on clean frames without performing denoising, creating redundant operations. In contrast, our method performs KV caching directly on the noisy latent as shown in Figure 1, which simultaneously serves two purposes: providing conditioning for subsequent chunks and progressing the denoising process to yield clean latents. This elimination of redundant computations directly translates to reduced latency and increased throughput. You can find more details in the following code from https://github.com/guandeh17/Self-Forcing/blob/main/pipeline/causal_inference.py:
>
> Lines 226-235: `# Step 3.3: rerun with timestep zero to update KV cache using clean context`
>
> This step in Self-Forcing incurs additional computational overhead, which our method avoids since we condition on the last noisy state rather than clean frames.

---

> > ### Author Response · Authors · 2025-12-04
> >
> > **3. Optimized Attention Window Size:** Our rolling KV cache mechanism, inspired by Self-Forcing, employs a more efficient context window strategy. While maintaining seamless rolling forward to avoid sliding-window overhead, we reduce the KV cache size from 6 chunks (used in the original Self-Forcing) to 4 chunks. This optimization is supported by our scaling analysis in Table 2, which shows that performance plateaus around window sizes of 12-27, with smaller windows providing better latency-memory trade-offs. The reduced window size contributes significantly to lower memory usage and **faster inference and lower in-flight latency** without compromising long-video generation quality.
> >
> > **Table 4: KV Cache Scaling Analysis**
> >
> > | Attention Window Size | Total Score | In-Flight Latency (s) | Memory (GB) |
> > |-----------------------|-------------|------------------------|-------------|
> > | 3                     | 80.9        | 0.37 ± 0.01            | 14.9        |
> > | 6                     | 81.3        | 0.38 ± 0.01            | 15.8        |
> > | 9                     | 82.3        | 0.40 ± 0.01            | 16.6        |
> > | 12                    | 84.3        | 0.46 ± 0.02            | 17.5        |
> > | 15                    | 84.2        | 0.51 ± 0.02            | 18.4        |
> > | 18                    | 84.4        | 0.54 ± 0.02            | 19.2        |
> > | 21                    | 84.5        | 0.59 ± 0.02            | 20.1        |
> > | 24                    | 84.3        | 0.64 ± 0.02            | 20.9        |
> > | 27                    | 84.5        | 0.68 ± 0.02            | 21.8        |
> >
> > **4. Efficient Tokenization Mechanism:** The tokenization process operates directly on the latent representations from the Tiny VAE [1], which encodes frames into tokens. As shown in Table 3, the Tiny VAE achieves substantially faster decoding with significantly fewer parameters than the Full VAE. For streaming applications, we further optimize the pipeline using an efficient Tiny VAE. By eliminating the VAE bottleneck, the Tiny VAE reduces decoding time by more than 10×, which also contributes to faster inference and **lower both first-frame latency and in-flight latency.**
> >
> > **Table 5: Comparison of Causal VAE Models**
> >
> > | Model               | Decoder Params | Compression Rate | Decoding Time(s) (81×832×480) |
> > |---------------------|----------------|------------------|-------------------------------|
> > | Full VAE (Wan 2.1)  | 73.3M          | 8×8×4            | 1.67                          |
> > | Tiny VAE (Wan 2.1) [1] | 9.84M       | 8×8×4            | 0.12                          |
> >
> > We hope this explanation clarifies your questions. Our code will be fully open-sourced upon the publication of our paper. You will be able to check the implementation details at that time.
> >
> > [1] https://github.com/madebyollin/taehv/

---

### Author Response · Authors · 2025-11-23
**Acknowledging Valuable Feedback and General Response to Reviewers**

We sincerely thank all reviewers for their insightful feedback and constructive suggestions. We are particularly gratified that the reviewers recognized the **conceptual elegance** of our simple but **appealing** diagonal denoising idea, which effectively bridges autoregressive conditioning with step-efficient diffusion distillation (Reviewer zYkc, Reviewer UG2y), its **strong practical impact for real-time streaming applications** (Reviewer i62A, Reviewer UG2y)—enabling the generation of a **5-second video in just 2.61 seconds (up to 31 FPS)**, which represents a 277.3× speedup over the undistilled model and doubles the acceleration ratio of the state-of-the-art (140×) without sacrificing visual quality—and the **well-motivated novelty** of our core technical contributions as well as the solid ablation studies that convincingly demonstrate the contribution of Diagonal Forcing and Flow Distribution Matching to temporal coherence and motion fidelity. (Reviewer uXB6, Reviewer zYkc).

In response to the valuable comments and concerns about details, we have conducted additional experiments, including **quantitative and fine-grained ablation studies**, to **rigorously validate the contribution of each proposed module**, and confirm that our parameter choices are grounded in **comprehensive experimentation**. In our rebuttal, we primarily address the reviewers' questions through **detailed experiments and systematic analysis** from the following aspects:

-   **Extended Acceleration Analysis (Reviewer zYkc, Reviewer UG2y, Reviewer uXB6):**
    1.  An Extended Acceleration Analysis deconstructs the synergistic optimizations of Diagonal Forcing (fewer denoising steps, efficient KV caching, optimized attention window, efficient tokenization), validated by controlled experiments under identical NFE budgets.
    2.  A Step Allocation Study explores strategies beyond our baseline, confirming the optimality of monotonically decreasing schedules and offering practical guidelines for different latency-quality trade-offs.
    3.  A Streaming Protocol Analysis details implementation specifics like chunk size selection, KV cache configuration, VAE tokenization, and pre-fill cost optimization, demonstrating the robustness of our O(TL) architecture for long sequences. **These additions provide substantial technical depth and empirical validation for our efficiency claims.**

-   **Quantitative Evaluation of Long-Range Performance (Reviewer i62A, Reviewer UG2y):** To further demonstrate the effectiveness of our Diagonal Forcing mechanism in mitigating exposure bias, we have generated and evaluated videos of extended duration (from 45s to up to **5 minutes**). The results, available at the bottom of (https://diagonal-distillation.github.io/), **show almost no significant quality degradation compared to shorter sequences**, providing strong evidence for our core claim of reduced error accumulation in long-range generation.

-   **Direct Measurement of Exposure Bias Mitigation (Reviewer UG2y):** We have introduced a new quantitative analysis to directly validate the training-inference alignment achieved by our Diagonal Forcing technique. By tracking metrics like saturation and contrast drift over long sequences (https://diagonal-distillation.github.io/image/contrast_ratio_and_saturability.png), we **provide concrete evidence** supporting our novel approach to addressing temporal consistency in streaming generation.

-   **Detailed Explanation of Flow Feature Extractor and Ablation Studies (Reviewer uXB6, Reviewer UG2y):** We have provided a detailed description of our flow feature extractor's architecture, supported by comprehensive ablation studies that systematically evaluate the impact of different designs and clarify the novelty of our design. This work **empirically validates the rationale behind our chosen configuration**. We have also supplemented the effectiveness of our method on models with larger parameter counts, different resolutions, and varying motion amplitudes.

For our revised paper and supplementary materials, to fully address the reviewers' requests for more details and rigor, we have significantly expanded our revised manuscript and supplementary materials, these additions include training configuration and optimization details (Appendix C), comparisons of temporal training strategies and model architecture specifics (Appendix D), pseudo-code for Diagonal Denoising with Noisy KV Cache (Appendix E), acceleration and step allocation analysis (Appendix F), technical details of the stream protocol (Appendix G), a detailed ablation study on motion flow field representation (Appendix H), and a scalability assessment on the Wan 2.1 14B model (Appendix I). We are grateful for the reviewers' insightful comments on the additional details, which have helped us clarify the novelty of our overall design.

---

> ### Author Response · Authors · 2025-12-04
>
> We hope that these revisions will help strengthen the presentation of our contributions and provide the VideoGen/World Model community with a well-validated framework for efficient interactive video generation. The encouraging feedback from the reviewers on our core ideas motivated us to make dedicated efforts in this revised version to address their concerns. We thank the reviewers again for their time and valuable input.
>
> Best regards,
> Authors of ICLR 2026 Conference Paper 18692

---

### Meta-Review · Area_Chair_bVUa · 2026-01-08

**Summary:**

All reviewers acknowledge that the Diagonal Distillation is a novel method for streaming video generation, but raised several concerns regarding its technical depth and clarity. Primary issues included missing experiments and technical details - reviewers demanded more rigorous implementation details regarding the motion flow estimator, the streaming protocol (chunk sizes and memory footprint), and the stability of the model over very long horizons (e.g., minutes). Finally, reviewer uXB6 asked for fair comparisons by benchmarking the method against baselines using matched computational budgets (Noise Function Evaluations or NFEs).

Overall, the paper is a solid contribution. The authors included a very comprehensive rebuttal with most concerns addressed, Hence, I am voting to accept the paper.

**Reviewer Concerns:**

The authors successfully addressed the majority of the reviewers' concerns through a very comprehensive rebuttal. They provided a detailed breakdown of the motion feature extractor, explaining its EMA-based MLP implementation on latents. They also provided new experiments - NFE table with matched compute budgets, proving their method is inherently more efficient than Self-Forcing, and results on the 14B parameter model to prove generalization. Evaluation on higer resolutions such as 720p would have made the paper even stronger.

**Reviewer Scores:**

Reviewer zYkc will maintain their 8, as their concerns were minor and the authors addressed cosmetic issues. Reviewer i62A would likely increase to a 7 as the authors provided the requested 5-minute video stability results and a thorough quantitative analysis of step-allocation heuristics.
Reviewer UG2y and uXB6, who were leaning towards rejection might increase their scores too, as authors provided most of the requested experiments and clarification.

---

### Decision · Program_Chairs · 2026-01-26

Accept (Poster)